# HBO1-MLL interaction promotes AF4/ENL/P-TEFb-mediated leukemogenesis

**Satoshi Takahashi[1,2], Akinori Kanai[3], Hiroshi Okuda[1], Ryo Miyamoto[1], Yosuke Komata[1], Takeshi Kawamura[4], Hirotaka Matsui[5], Toshiya Inaba[3], Akifumi Takaori-Kondo[2], Akihiko Yokoyama[1,2,6]***

[1]Tsuruoka Metabolomics Laboratory, National Cancer Center, Tsuruoka, Japan; [2]Department of Hematology and Oncology, Kyoto University Graduate School of Medicine, Kyoto, Japan; [3]Department of Molecular Oncology and Leukemia Program Project, Research Institute for Radiation Biology and Medicine, Hiroshima University, Hiroshima, Japan; [4]Isotope Science Center, The University of Tokyo, Tokyo, Japan; [5]Department of Molecular Laboratory Medicine, Faculty of Life Sciences, Kumamoto University, Kumamoto, Japan; [6]Division of Hematological Malignancy, National Cancer Center Research Institute, Tokyo, Japan

**Abstract** Leukemic oncoproteins cause uncontrolled self-renewal of hematopoietic progenitors by aberrant gene activation, eventually causing leukemia. However, the molecular mechanism underlying aberrant gene activation remains elusive. Here, we showed that leukemic MLL fusion proteins associate with the HBO1 histone acetyltransferase (HAT) complex through their trithorax homology domain 2 (THD2) in various human cell lines. MLL proteins associated with the HBO1 complex through multiple contacts mediated mainly by the ING4/5 and PHF16 subunits in a chromatin-bound context where histone H3 lysine 4 tri-methylation marks were present. Of the many MLL fusions, MLL-ELL particularly depended on the THD2-mediated association with the HBO1 complex for leukemic transformation. The C-terminal portion of ELL provided a binding platform for multiple factors including AF4, EAF1, and p53. MLL-ELL activated gene expression in murine hematopoietic progenitors by loading an AF4/ENL/P-TEFb (AEP) complex onto the target promoters wherein the HBO1 complex promoted the association with AEP complex over EAF1 and p53. Moreover, the NUP98-HBO1 fusion protein exerted its oncogenic properties via interaction with MLL but not its intrinsic HAT activity. Thus, the interaction between the HBO1 complex and MLL is an important nexus in leukemic transformation, which may serve as a therapeutic target for drug development.

*For correspondence: ayokoyam@ncc-tmc.jp

## Introduction

Mutated transcriptional regulators often cause uncontrolled self-renewal of immature hematopoietic precursors, which leads to aggressive leukemia. MLL (also known as KMT2A) is a transcriptional maintenance factor that upregulates homeobox (*HOX*) genes in development (*Yu et al., 1998*). Chromosomal translocations generate fusion genes encoding various MLL fusion proteins with more than 80 different partners to induce leukemia (*Meyer et al., 2018*). MLL fusion proteins cause uncontrolled self-renewal by constitutively activating various oncogenic genes (e.g., *HOXA9*, *MEIS1*), whose expression is normally restricted in immature precursors such as hematopoietic stem cells (HSCs) (*Krivtsov et al., 2006*). However, the mechanisms by which MLL fusion proteins aberrantly activate the expression of HSC-specific genes remain elusive.

MLL fusion proteins form a complex with MENIN (*Yokoyama et al., 2005*; *Yokoyama et al., 2004*), which leads to further association with LEGDF (*Yokoyama and Cleary, 2008*). MLL fusion

proteins bind to their target chromatin through the CXXC domain of MLL, which specifically recognizes unmethylated CpGs, and the PWWP domain of LEDGF, which selectively binds to di/tri-methylated histone H3 lysine 36 (H3K36me2/3) (*Ayton et al., 2004*; *Birke et al., 2002*; *Pradeepa et al., 2012*; *van Nuland et al., 2013*). The CXXC and PWWP domains constitute the minimum targeting module (MTM) which can stably bind to the human MLL target gene promoters (e.g., *HOXA9*) (*Okuda et al., 2014*). Because unmethylated CpGs and H3K36me2/3 marks are associated with transcriptional activation, MLL fusion proteins target a broad range of previously transcribed CpG-rich promoters. Although MLL fuses with a variety of partners, most MLL fusion-mediated leukemia cases are caused by the fusions with the AF4 family (e.g., AF4 also known as AFF1, AF5Q31 also known as AFF4) and ENL family (e.g., ENL also known as MLLT1, AF9 also known as MLLT3) (*Meyer et al., 2018*). AF4 family proteins form a biochemically stable complex with ENL family proteins and the P-TEFb elongation factor (composed of CDK9 and CyclinT1/2), which we termed AEP complex (as in AF4 family/ENL family/P-TEFb complex) (*Yokoyama et al., 2010*). AF4 family proteins recruit the SL1 complex (*Okuda et al., 2015*), which is composed of TBP and the TAF1A/B/C/D subunits and is known to initiate ribosomal RNA transcription by RNA polymerase I (*Goodfellow and Zomerdijk, 2013*). An artificial construct of MTM fused to the binding platform for SL1 activated the murine *Hoxa9* gene and transformed hematopoietic progenitors (HPCs) (*Okuda et al., 2015*), indicating that the AF4 family protein activates RNA polymerase II (RNAP2)-dependent transcription, presumably by loading TBP onto the target promoters via SL1. However, why MLL fusion proteins preferentially use the AEP/SL1-mediated transcription pathway is unclear.

In this study, we identified the evolutionarily conserved trithorax homology domain 2 (THD2) of MLL (previously termed TRX2 domain) as a key structure required for aberrant self-renewal mediated by MLL fusion proteins. A subsequent proteomic approach identified the HBO1 (also known as KAT7 and MYST2) histone acetyltransferase (HAT) complex as an associating factor for THD2. HBO1 is a member of the MYST HAT family responsible for histone H3 lysine 14 acetylation (H3K14ac) when complexed with BRPF family proteins, while it mainly acetylates histone H4 lysine 5/8/12 when complexed with JADE family proteins (*Lalonde et al., 2013*; *Mishima et al., 2011*). HBO1 was recently identified as a therapeutic vulnerability of leukemia stem cells by genetic screening (*Au et al., 2020*; *MacPherson et al., 2020*). However, its molecular functions on leukemic proteins remain largely elusive. Our detailed structure/function analysis demonstrated that HBO1-MLL interaction promotes AEP-dependent gene activation in MLL fusion-mediated leukemic transformation. Moreover, another leukemic fusion of nucleoporin-98 (NUP98) and HBO1(i.e., NUP98-HBO1) also transformed HPCs via association with MLL. Hence, we propose that HBO1-MLL interaction modules can be utilized as molecular targets for developing drugs that specifically dismantle the oncogenic transcriptional machinery.

## Results

### THD2-mediated functions promote MLL fusion-dependent leukemic transformation

MLL fusion proteins confer unlimited self-renewal ability, leading to the immortalization of HPCs (*Lavau et al., 1997*). MLL-ELL is among the frequently observed MLL fusions associated with acute myelogenous leukemia (*Meyer et al., 2018*). To determine the domain structures required for MLL-ELL-mediated leukemic transformation, we performed myeloid progenitor transformation assays (*Lavau et al., 1997*; *Okuda and Yokoyama, 2017b*), wherein murine HPCs were retrovirally transduced with an MLL fusion gene and cultured in a semisolid medium supplemented with myeloid cytokines (*Figure 1A*). MLL-ELL transformed HPCs, as previously reported (*DiMartino et al., 2000*; *Luo et al., 2001*), featured vigorous colony-forming capacities at the third and fourth passages, and showed elevated *Hoxa9* expression at the first and second passages. A deletion mutant lacking THD2 failed to transform (*Figure 1A* and *Figure 1—figure supplement 1A*), underscoring the biological significance of THD2. A minimalistic artificial construct of HA-tagged MTM fused to an intact Occludin homology domain (OHD) of ELL failed to transform HPCs (*Figure 1A*, see MTMh-ELL″), whereas inclusion of THD2 to MTM (hereafter denoted as MTMT) caused constitutive activation of *Hoxa9* and immortalization of HPCs (*Figure 1A*, see MTMTh-ELL″). These results indicate that MLL-ELL transforms HPCs through THD2-mediated functions.

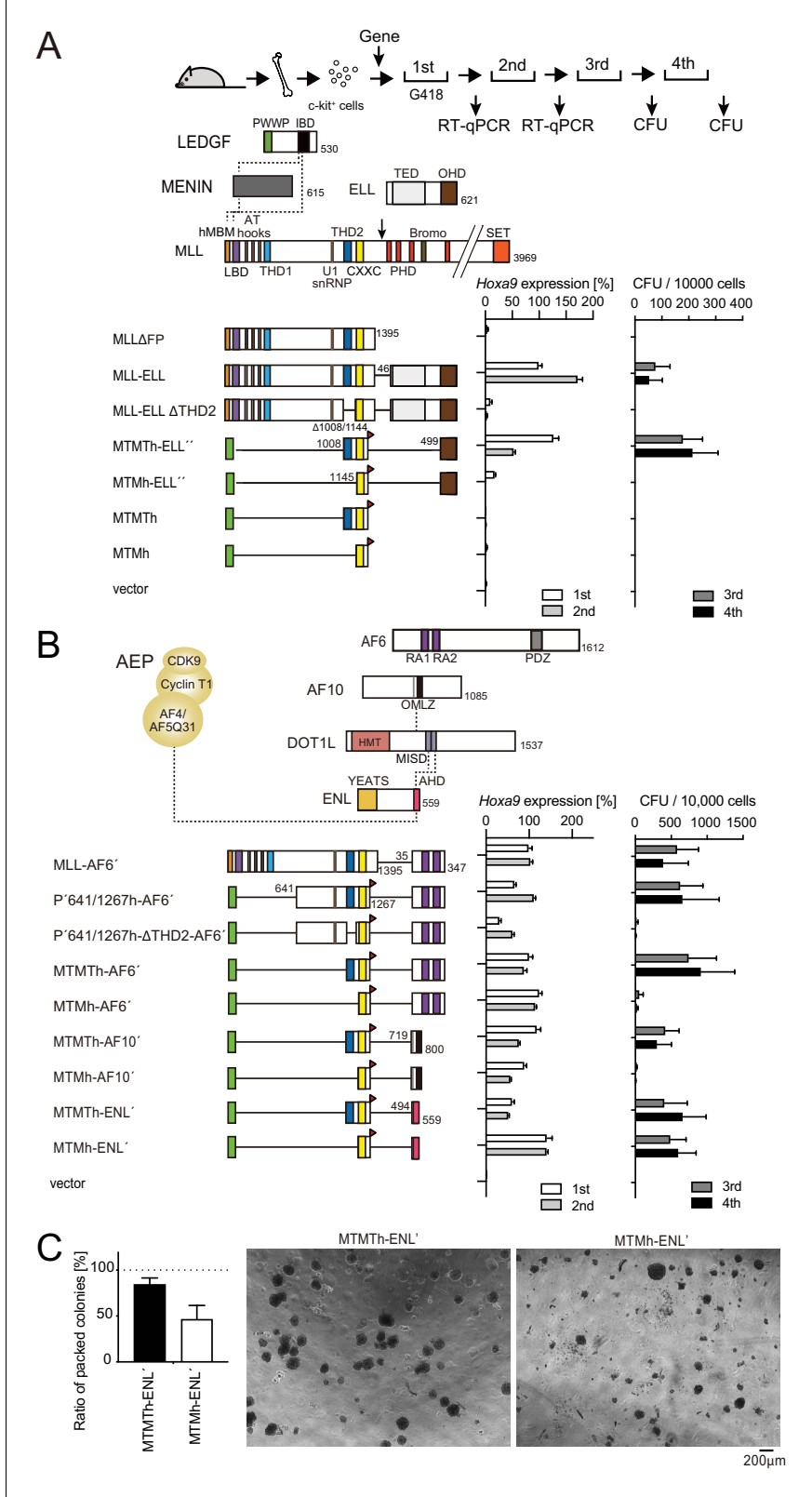

**Figure 1.** Trithorax homology domain 2 (THD2)-mediated functions promote MLL-fusion-dependent leukemic transformation. (**A**) Structure/function analysis of MLL-ELL. Various MLL-ELL constructs were examined for the transformation of myeloid progenitors. HA-tag (h: indicated as a red triangle) was fused to MTM and MTMT constructs. A schema of myeloid progenitor transformation assay is shown on top. *Hoxa9* expression normalized to

*Figure 1 continued*

*Gapdh* in first-round and second-round colonies (left) is shown as the relative value of MLL-ELL (arbitrarily set at 100%) with error bars (mean ± SD of PCR triplicates). Colony-forming ability at third- and fourth-round passages (right) is shown with error bars (mean ± SD of ≥3 biological replicates). IBD: integrase-binding domain; hMBM: high-affinity MENIN-binding motif; LBD: LEDGF-binding domain; THD1/2: trithorax homology domains 1 and 2; SET: Suver3-9/enhancer-of-zeste/trithorax motif; TED: transcription elongation domain; OHD: occludin homology domain. (B) Requirement of THD2 for various MLL fusion proteins in leukemic transformation. Various MLL fusion constructs were examined for the transformation of myeloid progenitors as in (A). RA1/2: RAS association domains 1/2; PDZ: PSD-95/Dlg/ZO-1 domain; OM: octapeptide motif; LZ: leucine zipper. MISD: minimum interaction site of DOT1L: YEATS: Yaf9 ENL AF9 Taf14 Sas5 domain; AHD: ANC1 homology domain. (C) Colony morphologies of MTMTh or MTMh-ENL′ transformed cells. The colonies on day 5 of fourth passage are shown with a scale bar. The ratio of compact colonies (≥100 total colonies were counted in each experiment) is shown on the left (mean ± SD of six biological replicates). Representative images are shown on the right.

The online version of this article includes the following figure supplement(s) for figure 1:

**Figure supplement 1.** Trithorax homology domain 2 (THD2) promotes MLL fusion-mediated leukemogenesis.

---

Next, we tested the structural requirements of other MLL fusions (i.e., MLL-AF6, MLL-AF10, and MLL-ENL) (*Figure 1B* and *Figure 1—figure supplement 1A*). An MLL-AF6 fusion construct, in which MLL is fused to the RA1 and RA2 domains of AF6 (see MLL-AF6′), fully transformed HPCs as previously reported (*Liedtke et al., 2010*). The minimalistic MTM-AF6 fusion construct (*Figure 1B*, see MTMh-AF6′) activated *Hoxa9* expression in the early passages but produced modest numbers of colonies in later passages, while the inclusion of THD2 (*Figure 1B*, see MTMTh-AF6′) conferred much more vigorous transforming capacities. Deletion of THD2 from a PWWP-MLL-AF6 fusion construct containing the residues 641/1267 of MLL (P′641/1267h-AF6′) abrogated its transforming ability, suggesting that MLL-AF6 requires THD2 to exert its full transforming potential (*Figure 1B* and *Figure 1—figure supplement 1A*). Nonetheless, an MLL-AF6′ construct lacking THD2 immortalized HPCs albeit less efficiently compared to MLL-AF6′ (*Figure 1—figure supplement 1B*, see MLL-AF6′ ΔTHD2), suggestive of some compensatory functions mediated by other MLL structures retained in MLL-AF6′ but missing in P′641/1267h-AF6′ (i.e., the residues 1/640 and 1268/1395). MLL-AF10 showed a similar trend, in line with our previous report (*Okuda et al., 2017*; *Figure 1B* and *Figure 1—figure supplement 1A,B*). In contrast, the oncogenic properties of MLL-ENL were not severely affected by the loss of THD2 in terms of colony-forming capacity (*Figure 1B* and *Figure 1—figure supplement 1B*; *Okuda et al., 2014*). However, the colony morphology of immortalized cells (ICs) transformed by the MTMh-ENL construct (MTMh-ENL′-ICs) was more differentiated compared to that of MTMTh-ENL′-ICs (*Figure 1C*), suggesting that THD2 is required to block differentiation to some extent. Leukemogenesis in vivo was compromised by the deletion of THD2 for MLL-AF6, -AF10, and -ENL (*Figure 1—figure supplement 1C*). These results indicate that MLL fusion proteins rely on THD2-mediated functions for leukemic transformation to varying degrees depending on their fusion partners. Among the MLL fusions tested, MLL-ELL most heavily relies on THD2-mediated functions.

## HBO1 complex associates with MLL proteins via THD2 at promoters

We previously showed that THD2 binds to AF4 family proteins (*Okuda et al., 2017*). Indeed, a FLAG-tagged MLL construct encompassing the residues 869/1152 efficiently co-precipitated with exogenously expressed AF4 or AF5Q31 (*Figure 2—figure supplement 1A*, see fMLL 869/1152 +37aa). However, deletion of THD2 from the FLAG-tagged MLLΔFP construct (containing the residues 1/1395: fMLL ΔFP) did not impair co-precipitation of AF4, indicating that the interaction with AF4 is not mediated by THD2. Sequencing analysis of the vector constructs revealed that the fMLL 869/1152+37aa construct contained an additional coding sequence tethered in frame for 37 residues derived from the pCMV5 vector, which corresponds to a part of the chorionic somatomammotropin hormone 1 gene (*Figure 2—figure supplement 1B*). Removal of the additional 37 residues by introducing a stop codon resulted in complete loss of association with AF4 family proteins (*Figure 2—figure supplement 1A*, see fMLL 869/1251). Moreover, a FLAG-tagged GAL4 fusion construct tethered to the additional 37 residues co-precipitated with AF5Q31 (see fGAL4+37aa). Thus, we concluded that our previous claim for THD2 as a binding platform for AF4 family proteins was false.

To identify bona fide associating factors for THD2, we purified two exogenously expressed THD2-containing proteins (i.e., fGAL4-MLL 869/1124 and MTMTh) from the chromatin fraction of HEK293T cells using the fractionation-assisted chromatin immunoprecipitation (fanChIP) method (*Miyamoto and Yokoyama, 2021*) and analyzed by mass spectrometry (*Figure 2A*). Components of the HBO1 complex including HBO1, PHF16 (also known as JADE3), MEAF6, and ING5 were detected in the purified materials (*Avvakumov et al., 2012*). Immunoprecipitation-western blotting (IP-WB) analysis confirmed that fGAL4-MLL 869/1124 co-precipitated with HBO1 complex components but not with exogenously expressed AF4 (*Figure 2B*). A series of MTMT fusion proteins were associated with HBO1 and PHF16, while the respective MTM fusion proteins were not (*Figure 2C*), confirming that THD2 mediates interaction with the HBO1 complex. Domain mapping analysis of MLL showed that the residues 869/1124 contain the major binding domains for the HBO1 complex (*Figure 2—figure supplement 2A*). Moreover, fGAL4-MLL 1052/1124 co-precipitated with the HBO1 complex components after DNaseI treatment, indicating that association of MLL and HBO1 is not mediated by DNA (*Figure 2—figure supplement 2B*). It should be noted that a small amount of HBO1 co-precipitated with the MLL proteins lacking the THD2 domain (*Figure 2—figure supplement 2C*, see fMLL ΔFP ΔTHD2 and fMLL-ENL ΔTHD2), suggesting that there is a secondary binding domain for the HBO1 complex outside of THD2, which may account for the moderate effects of THD2 deletion from the full-length MLL fusion constructs (*Figure 1—figure supplement 1B*).

Interaction between endogenous MLL proteins and HBO1 was confirmed in leukemia cell lines such as HB1119 and REH (*Figure 2—figure supplement 2D,E*). Chromatin immunoprecipitation (ChIP) followed by deep sequencing (ChIP-seq) of HB1119 cells, which endogenously express MLL-ENL, demonstrated that HBO1 complex components colocalized with MLL-ENL at the MLL target genes (e.g., *MYC*, *HOXA9*, and *MYB*) in a genome-wide manner (*Figure 2D,E* and *Figure 2—figure supplement 3A*). CRISPR/Cas9-mediated sgRNA competition assays demonstrated that *Kat7/KAT7* (the gene encoding HBO1) is required for the proliferation of various MLL fusion-ICs (*Figure 2F*) and in human leukemia cell lines (*Figure 2—figure supplement 3B*). These results are consistent with the recent reports, which showed that the HBO1 complex associates with multiple MLL fusion proteins and plays a critical role in the maintenance of leukemia stem cells (*Au et al., 2020*; *MacPherson et al., 2020*). Taken together, MLL fusion proteins associate with the HBO1 complex through THD2 at the target promoters.

## MLL recruits the HBO1, AEP, and SL1 complexes to promoters

To elucidate the functional relationship between wild-type MLL and the HBO1 complex, we examined the genomic localization of the HBO1 complex, wild-type MLL, and its related transcriptional regulators in HEK293T cells (*Figure 3A* and *Figure 3—figure supplement 1A*). MLL localizes at the transcription start sites (TSSs) of CpG-rich genes including *RPL13A*, *MYC*, and *CDKN2C* (*Okuda et al., 2017*). Distribution of the HBO1 complex was enriched at promoter proximal transcribed regions (0–2 kb from the TSS) (*Figure 3B*) consistent with a previous report (*Avvakumov et al., 2012*), suggesting its implication in transcription initiation/elongation. The HBO1 complex was localized at the MLL-occupied promoters in a genome-wide manner similarly to that in HB1119 cells. We previously showed that MLL associates with the MOZ complex and yet-to-be activated RNAP2 whose heptapeptide repeats are not phosphorylated (RNAP2 non-P) on target promoters, and that it colocalizes with various ENL-containing complexes including the AEP and DOT1L complexes (*Miyamoto et al., 2020*; *Okuda et al., 2017*). The ChIP signal intensities of MLL were highly correlated with those of HBO1, MOZ, and RNAP2 non-P (*Figure 3C*), supporting the functional interactions of MLL with those MLL-associated factors. The ChIP signals of AEP (e.g., AF4, ENL) and SL1 (e.g., TAF1C) are weakly correlated with that of MLL, presumably because AEP and SL1 are indirectly recruited to MLL target promoters in a context-dependent manner.

To examine the role of MLL in the genomic localization of the HBO1 complex, we analyzed MLL-deficient HEK293T cells by qRT-PCR and ChIP-qPCR. Depletion of MLL by the knockout of *KMT2A* (the gene encoding MLL) reduced the expression of *MYC* and *CDKN2C* (*Figure 3D,E*), as previously reported (*Miyamoto et al., 2020*), and caused a marked reduction in the ChIP signals of the HBO1 complex components (i.e., PHF16, MEAF6) at the *MYC* and *CDKN2C* loci (*Figure 3F* and *Figure 3—figure supplement 1B*). The HBO1 complex is presumed to target the chromatin containing histone H3 lysine 4 (H3K4me3) marks to which the PHD finger of the ING5 subunit specifically binds (*Champagne et al., 2008*). H3K4me3 marks were substantially reduced at the *MYC* and *CDKN2C*

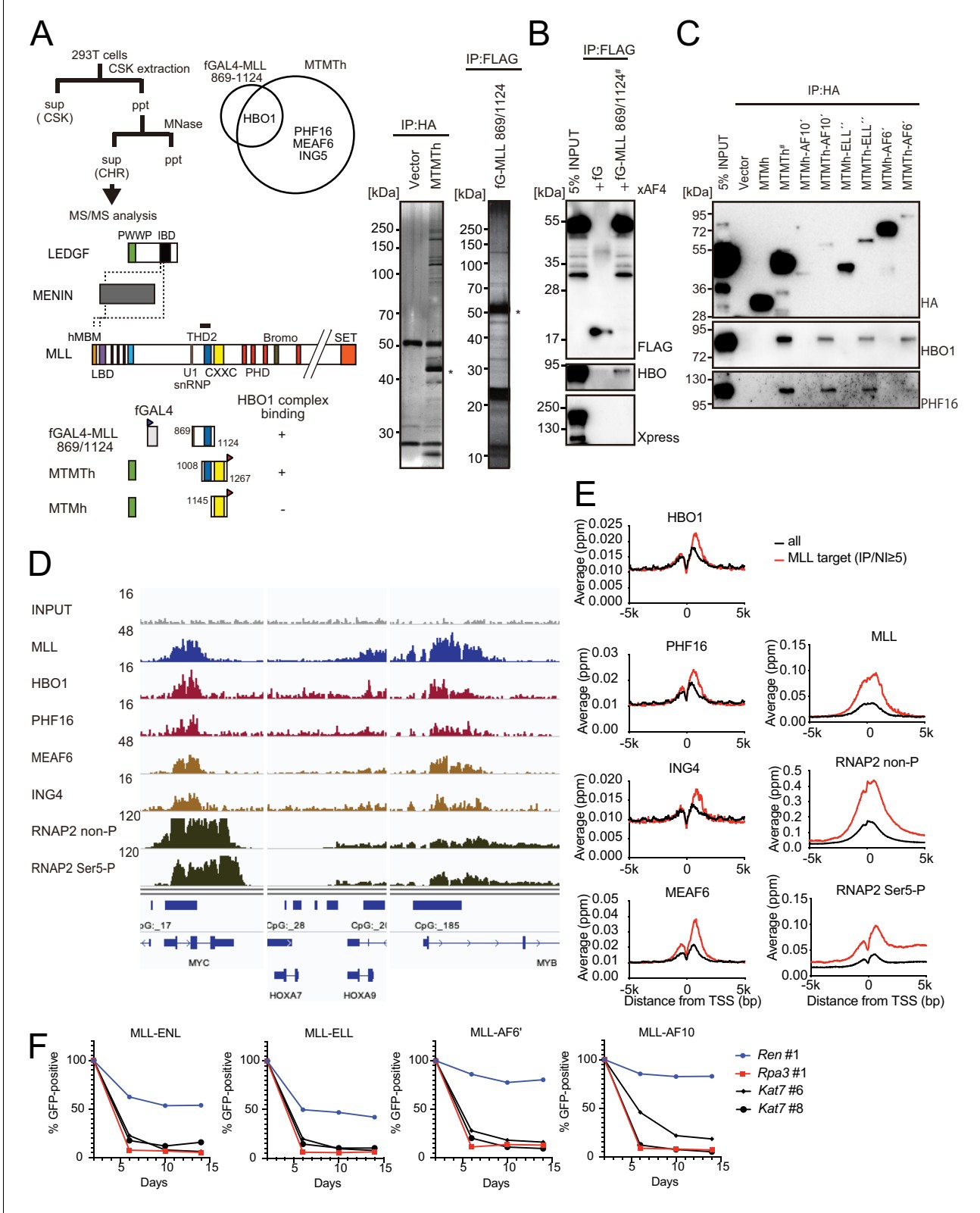

**Figure 2.** HBO1 complex associates with MLL proteins at promoters. (**A**) Purification of trithorax homology domain 2 (THD2) domain-associating factors. A FLAG-tagged (f: indicated as a blue triangle) GAL4 DNA-binding domain fused with the MLL fragment containing the residues 869–1124 or HA-tagged MTMT fragment was transiently expressed in HEK293T cells. A schema of the fractionation-assisted chromatin immunoprecipitation (fanChIP) method is shown on top. The transgene products were purified from the chromatin fraction and analyzed by mass spectrometry. Silver-stained

*Figure 2 continued on next page*

*Figure 2 continued*

images (right) of the purified materials are shown. Asterisk indicates the position of the transgene products. A Venn diagram of identified THD2 domain-associating factors by mass spectrometry is shown. (B) Association of GAL4-THD2 fusion with HBO1. Immunoprecipitation (IP)-western blotting of the chromatin fraction of HEK293T cells transiently expressing FLAG-tagged GAL4-MLL 869/1124 construct and Xpress-tagged AF4 (xAF4) was performed. Co-purification of HBO1, but not xAF4, was confirmed. #: the sample used for the input. (C) THD2-dependent association with the HBO1 complex. IP-western blotting of the chromatin fraction of virus-packaging cells, transiently expressing various HA-tagged MTMT (or MTM) fusion constructs, was performed. (D) Genomic localization of MLL and the HBO1 complex components in HB1119 cells. The ChIP followed by deep sequencing (ChIP-seq) profiles were visualized using the Integrative Genomics Viewer (The Broad Institute). The minimum value of the y-axis was set at 0, while the maximum value for each sample is indicated. (E) Average distribution of proteins near the transcription start sites (TSSs) of HB1119 cells. Genes whose MLL ChIP signal/input ratio at the promoter proximal transcribed region was $\geq 5$ were defined as MLL target genes. Average ChIP signal distribution of indicated proteins at the MLL target genes (red) or all genes (black) is shown. The y-axis indicates the frequency of the ChIP-seq tag count (ppm) in 25 bp increments. (F) Requirement of HBO1 for myeloid progenitors immortalized by various oncogenes. sgRNA competition assays for *Hbo1* were performed on immortalized myeloid progenitors. The ratio of GFP-positive cells co-expressing sgRNA was measured by flow cytometry. sgRNA for Renilla luciferase (*Ren*) was used as a negative control, which does not affect proliferation. sgRNA for *Rpa3* was used as a positive control, which impairs proliferation.

The online version of this article includes the following figure supplement(s) for figure 2:

**Figure supplement 1.** Trithorax homology domain 2 (THD2) does not mediate association with AF4 family proteins.
**Figure supplement 2.** HBO1 complex binds to MLL in various contexts.
**Figure supplement 3.** HBO1 is required for proliferation of MLL leukemia cells.

promoters as they were presumably provided by MLL in part. These results suggest that the decreased presence of the HBO1 complex may be due to the reduced H3K4me3 marks. Transient expression of a THD2-containing MLL construct (MTMTh) did not rescue the recruitment of the HBO1 complex onto the MYC promoter (*Figure 3—figure supplement 1C*), suggesting that H3K4 methylation is a prerequisite for the recruitment of the HBO1 complex. In addition, the ChIP signals of AEP/SL1 components (i.e., ENL, CCNT1, AF4, and TAF1C) were also reduced at these loci (*Figure 3F* and *Figure 3—figure supplement 1B*). These results indicate that MLL recruits the HBO1, AEP, and SL1 complexes to the MYC and CDKN2C promoters in part via H3K4 methylation.

## MLL-ELL transforms through the common binding platform for AF4 and EAF1

Because MLL-ELL is highly dependent on THD2-mediated HBO interaction, we next dissected the function of the ELL portion in MLL-ELL-mediated leukemic transformation. ELL has a transcriptional elongation activity and is associated with a variety of proteins including AF4 and EAF family proteins (*Lin et al., 2010*; *Shilatifard et al., 1996*; *Simone et al., 2003*; *Simone et al., 2001*). Myeloid progenitor transformation assays demonstrated that an intact OHD, which is responsible for the association with both AF4 and EAF1 (*Lin et al., 2010*; *Simone et al., 2001*), is required for transformation as previously reported (*Figure 4A,B* and *Figure 4—figure supplement 1A*; *DiMartino et al., 2000*; *Luo et al., 2001*). ChIP-qPCR analysis of FLAG-tagged GAL4 constructs fused to ELL confirmed that OHD is responsible for the recruitment of both AF4 and EAF1 (*Figure 4—figure supplement 1B*). Deletion of OHD resulted in the loss of interaction with both AF4 and EAF1 family proteins, which is correlated with the transforming properties (*Figure 4B* and *Figure 4—figure supplement 1C*). It should be noted that several processed forms of AF4 (e.g., 110 kDa) were observed in the co-precipitates, whose amounts were specifically increased in the chromatin fraction of HEK293T cells transiently expressing GAL4-ELL proteins harboring an intact OHD (*Figure 4B* and *Figure 4—figure supplement 1C*), suggesting that ELL induces the processing of AF4 and tethers both processed and unprocessed forms of AF4 to the chromatin. ELL-AF4 interaction was attenuated by co-expression of EAF1, whereas ELL-EAF1 interaction was augmented by co-expression of AF4 (*Figure 4C,D*), suggesting that these interactions occur sequentially but not simultaneously, wherein the ELL-AF4 interaction precedes the ELL-EAF1 interaction (*Figure 4E*). Taken together, MLL-ELL exerts its transforming properties through the common binding platform for AF4 and EAF1.

## AF4 and EAF1 form two distinct SL1/MED26-containing complexes

AF4 and EAF1 family proteins share some structural similarities; both have the SDE motif enriched with serine, aspartic acid, and glutamic acid, and the DLXLS motif whose consensus sequence is

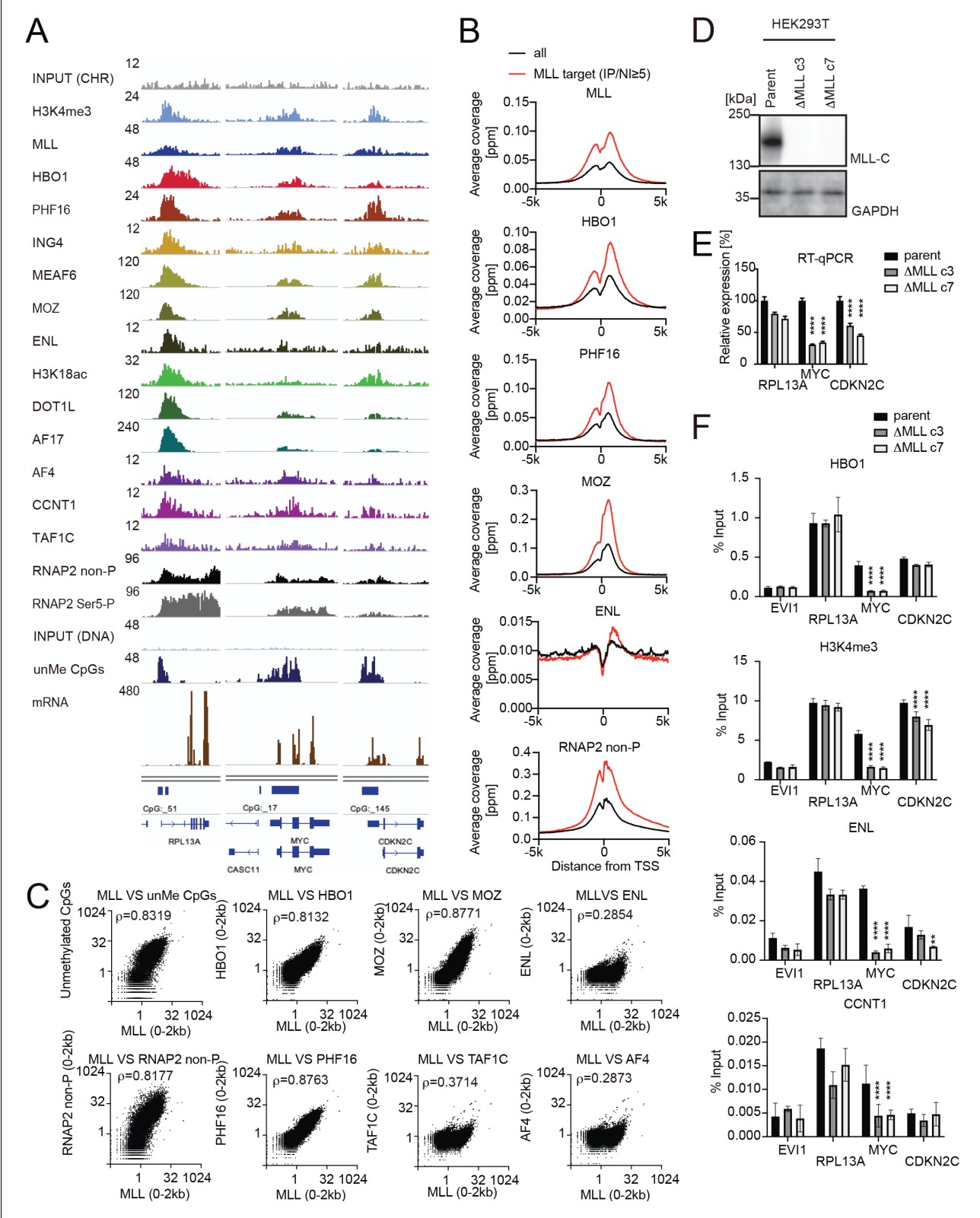

**Figure 3.** MLL recruits the HBO1 and AEP/SL1 complexes to promoters. (**A**) Genomic localization of various transcriptional regulators/epigenetic marks in HEK293T cells. Chromatin immunoprecipitation (ChIP) followed by deep sequencing (ChIP-seq) analysis was performed on the chromatin of HEK293T cells for the indicated proteins/modifications. CIRA-seq data for unmethylated CpGs (unMe CpGs) and RNA-seq data are shown for comparison. (**B**) Average distribution of proteins near transcription start sites (TSSs) of HEK293T cells. Average ChIP signal distribution of indicated proteins at the MLL

*Figure 3 continued on next page*

*Figure 3 continued*

target genes (red) or all genes (black) is shown as in *Figure 2E*. (**C**) Relative occupation by multiple factors at all TSSs. ChIP-seq or CIRA-seq tags of indicated proteins at all genes were clustered into a 2 kb bin (0 to +2 kb from the TSS) and are presented as XY scatter plots. Spearman's rank correlation coefficient (ρ) is shown. (**D**) Expression of MLL in MLL-deficient HEK293T cell lines. (**E**) Expression of MLL target genes in two independently generated MLL-deficient clones. Relative expression levels normalized to *GAPDH* are shown with error bars (mean ± SD of PCR triplicates) by RT-qPCR. Data are redundant with our previous report (*Miyamoto et al., 2020*). (**F**) Localization of MLL, the HBO1 complex, the AEP complex, and the SL1 complex in two independently generated MLL-deficient clones. ChIP-qPCR was performed for indicated genes using qPCR probes designed for the TSS of each gene. ChIP signals are expressed as the percent input with error bars (mean ± SD of biological triplicates). Statistical analysis was performed by ordinary two-way ANOVA comparing each sample with the parent cells. *p ≤ 0.05, **p ≤ 0.01, ***p ≤ 0.001, ****p ≤ 0.0001. The online version of this article includes the following figure supplement(s) for figure 3:

**Figure supplement 1.** MLL colocalizes with the HBO1 and AF4/ENL/P-TEFb (AEP)/SL1 complexes at promoters containing H3K4me3 marks.

LXXDLXLS (*Figure 5A,B*). The NKW motif was found in the AF4 family but not in the EAF family. The SDE motif of AF4 associates with the SL1 complex and its DLXLS motif associates with MED26 (*Okuda et al., 2015*; *Okuda et al., 2016*; *Takahashi et al., 2011*). IP-WB analysis demonstrated that EAF1 was also associated with SL1 and MED26 through its C-terminal domain containing the SDE and DLXLS motifs (*Figure 5C*, see fGAL4-EAF1-C). ChIP-qPCR analysis confirmed that the C-terminal portion of EAF1 recruited TAF1C and MED26 to the GAL4-responsive promoter (*Figure 5D*, see fGAL4-EAF1-C). Nevertheless, the GAL4-ELL fusion failed to associate with or recruit TAF1C and MED26 (*Figure 5C,D*, see fGAL4-ELL'), indicating that ELL-bound EAF1 or AF4 is unable to interact with TAF1C and MED26, and therefore must dissociate from ELL to form a complex with SL1 and MED26 (*Figure 5E,F*). It should be noted that GAL4-EAF1 failed to pull down endogenous AF4 and ENL (*Figure 5C,D*, see fGAL4-EAF1), while a GAL4-AF4 protein containing an ELL-binding domain (ALF) also failed to pull down exogenously expressed EAF1 (*Figure 5C,D*, see fGAL4-AF4-2N and C), indicating that the AF4/ELL/EAF1 trimer complex is unstable and that ELL mostly binds AF4 or EAF family proteins in a mutually exclusive manner.

In addition, GAL4-EAF1 co-precipitated with exogenously expressed EAF1 through the EAF family homology domain (EHD), indicating that EAF1 forms a homodimer (*Figure 5C*, see the HA blot of exogenously expressed HA-tagged EAF1). It has been shown that EAF1 binds to ELL through two distinct contacts (*Simone et al., 2003*). Accordingly, both N- and C-terminal halves of EAF1 co-precipitated with endogenous ELL (*Figure 5C*, see fGAL4-EAF1-N and -C in the upper ELL blot). However, when EAF1 was overexpressed, co-precipitation of endogenous ELL by the C-terminal half of EAF1 was no longer detected (*Figure 5C*, see fGAL4-EAF1-C in the lower ELL blot), suggesting that free ELL preferentially binds to the EAF1 dimer over the monomeric C-terminal half of EAF1. GAL4-ELL co-precipitated with endogenous ELL presumably mediated by an EAF1 dimer (*Figure 5C*, see fGAL4-ELL' in the upper ELL blot), whereas it failed to pull down ELL when exogenous ELL was overexpressed to absorb free EAF1 dimers (*Figure 5C*, see fGAL4-ELL' in the HA blot of exogenously expressed HA-tagged ELL). Taken together, these results suggest that ELL forms at least two different stable complexes, one is with AF4 family proteins, which subsequently leads to the formation of an AEP/SL1/MED26 complex (*Figure 5E*), and the other is an EAF1/ELL homodimer complex, which leads to the formation of an EAF1/SL1/MED26 complex (*Figure 5F*). These two SL1/MED26 containing complexes are similar in composition but different in three key functions. The AEP/SL1/MED26 complex has the NKW motif which is required for AF4-dependent gene activation (*Okuda et al., 2015*), and contains P-TEFb which promotes transcription elongation, and ENL family proteins which tether AEP on acetylated chromatin (*Erb et al., 2017*; *Li et al., 2014*; *Wan et al., 2017*). Thus, we presumed that the AEP/SL1/MED26 complex is competent for transactivation whereas the EAF1/SL1/MED26 complex is not.

## MLL-ELL-mediated leukemic transformation is driven by interaction with AEP, while MLL-EAF1-mediated transformation is promoted by homodimerization

Previously, *Luo et al., 2001*, demonstrated that an artificial fusion construct of MLL and EAF1 transformed HPCs and induced leukemia, which suggested that the ELL-EAF1 interaction, rather than ELL-AEP interaction, played a major role in MLL-ELL-mediated leukemic transformation. To examine the structural requirements for MLL-EAF1-mediated transformation, we generated artificial fusion

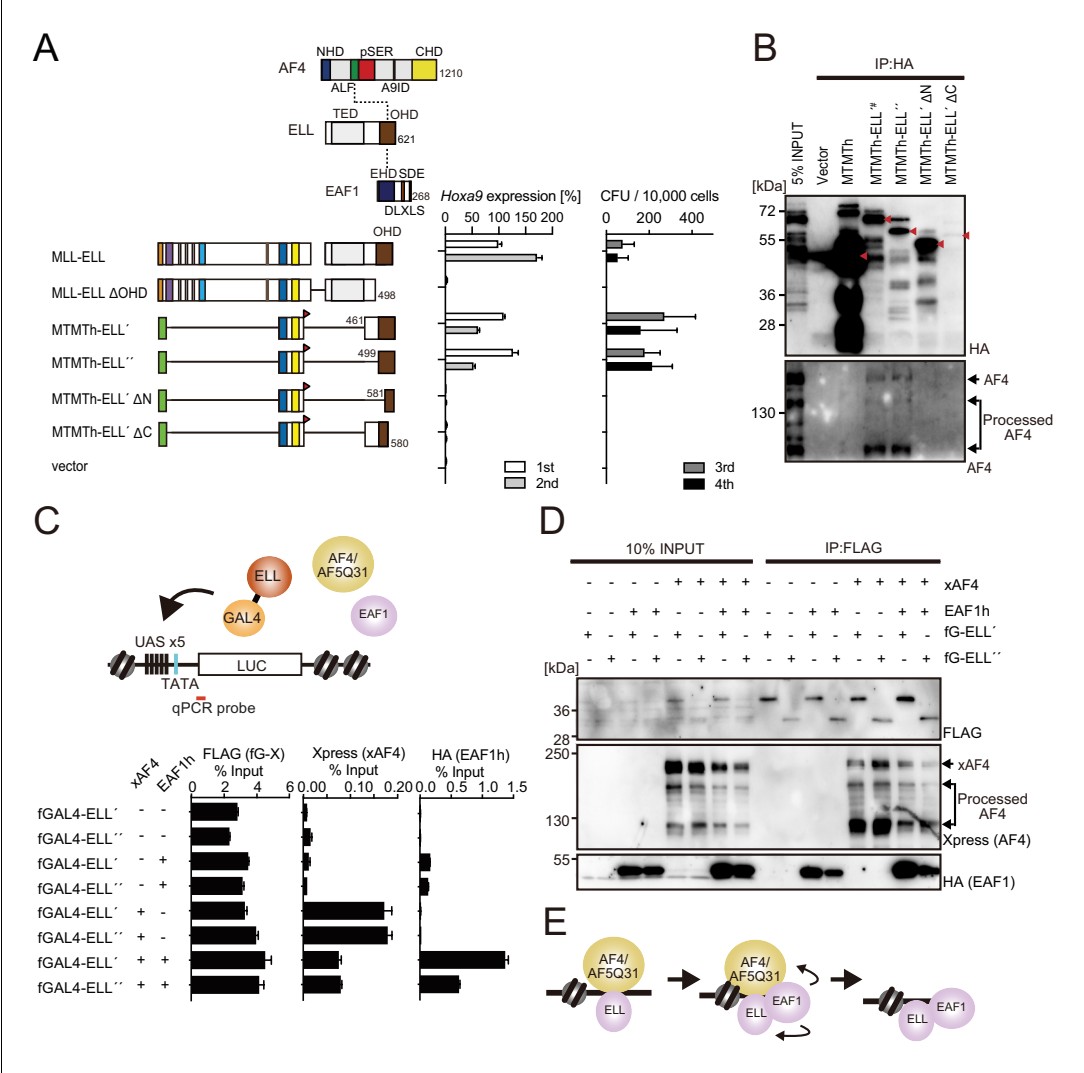

**Figure 4.** MLL-ELL transforms through the common binding platform for AF4 and EAF1. (**A**) Requirement of the occludin homology domain (OHD) in MLL-ELL-mediated transformation. Various MLL-ELL constructs carrying mutations in the ELL portion were examined for the transformation of myeloid progenitors, as in *Figure 1A*. NHD: N-terminal homology domain; ALF: AF4/LAF4/FMR2 homology domain; pSER: poly-serine; A9ID: AF9 interaction domain; CHD: C-terminal homology domain; EHD: EAF family homology domain; DLXLS: DLXLS motif; SDE: SDE motif. (**B**) OHD-dependent association with AF4. Immunoprecipitation (IP)-western blotting of the chromatin fraction of virus-packaging cells, transiently expressing various HA-tagged MTMT-ELL fusion constructs, was performed. Endogenous AF4 proteins co-purified with MTMTh-ELL proteins were visualized by an anti-AF4 antibody. (**C**) Sequential recruitment of AF4 and EAF1 by ELL. HEK293TL cells (*Okuda et al., 2015*), which harbor GAL4-responsive reporter, were transfected with various combinations of FLAG-tagged GAL4 fusion proteins, xAF4, and HA-tagged EAF1 (EAF1h), and were subjected to chromatin immunoprecipitation (ChIP)-qPCR analysis. A qPCR probe near the GAL4-responsive elements (UAS) was used. The ChIP signals were expressed as the percent input with error bars (mean ± SD of PCR triplicates). TATA: TATA box; LUC: luciferase. (**D**) Effects of overexpression of AF4 or EAF1 on ELL complex formation. IP-western blotting of the chromatin fraction of HEK293TL cells, transiently expressing various combinations of FLAG-tagged GAL4-ELL proteins, xAF4, and EAF1h, was performed. (**E**) A model of the sequential association between ELL, AF4 family proteins, and EAF family proteins. The online version of this article includes the following figure supplement(s) for figure 4:

**Figure supplement 1.** Functions of the ELL portion.

constructs in which MTM or MTMT is fused to EAF1 domains and examined their transforming properties (*Figure 5G* and *Figure 5—figure supplement 1A*). As reported previously (*Okuda et al., 2015*), an MTM construct fused to an AF4 portion containing the SDE and NKW motifs (i.e., MTMh-AF4-2C-abc) activated *Hoxa9* expression and immortalized HPCs, while a similar construct lacking the NKW motif (i.e., MTMh-AF4-2C-ab) failed to transform (*Figure 5G*). Accordingly, an MTM construct fused to the C-terminal half of EAF1 containing the SDE motif but lacking the NKW motif

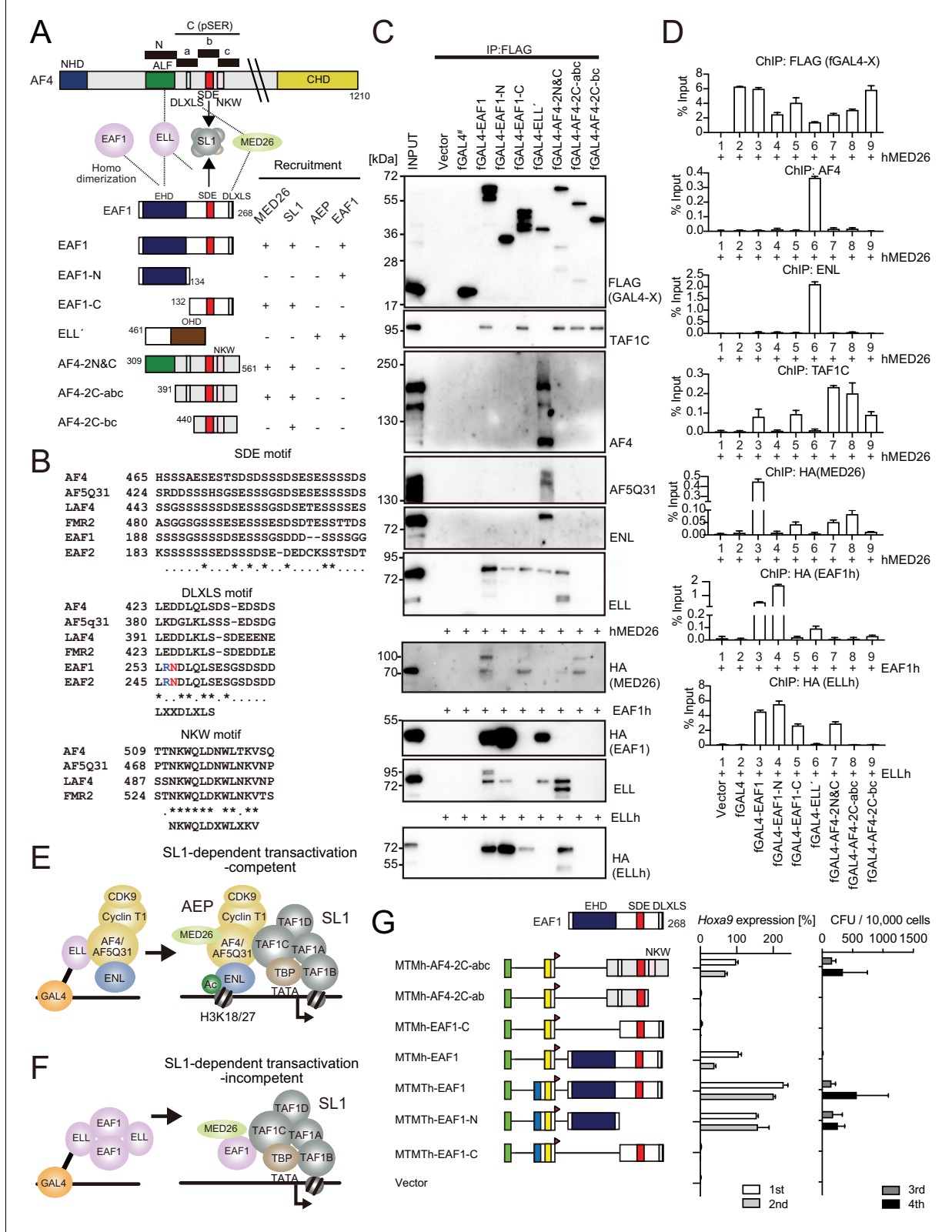

**Figure 5.** AF4 and EAF1 form distinct SL1/MED26-containing complexes. (**A**) A schema of the structures of AF4, ELL, and EAF1. ALF is responsible for interaction with ELL, pSER is responsible for interaction with SL1, and DLXLS is responsible for interaction with MED26. EAF family homology domain (EHD) is responsible for homodimerization and interaction with ELL. NKW:NKW motif responsible for transcriptional activation. (**B**) Alignment of the amino acid sequences of the SDE, DLXLS, and NKW motifs of human AF4 family proteins and EAF family proteins. Conserved residues are

*Figure 5 continued on next page*

*Figure 5 continued*

indicated by asterisks. (C) Association of EAF1, ELL, and AF4 domains with various associating factors on chromatin. Immunoprecipitation (IP)-western blotting of the chromatin fraction of HEK293TL cells transiently expressing various FLAG-tagged GAL4 fusion proteins, with or without indicated HA-tagged constructs, was performed as in *Figure 4D*. Endogenous proteins were detected by specific antibodies for each protein, while exogenous proteins were detected by antibodies for FLAG or HA tag. (D) Recruitment of various transcriptional regulatory proteins by EAF1, ELL, and AF4 domains. Chromatin immunoprecipitation (ChIP)-qPCR analysis of HEK293TL cells transiently expressing various combinations of FLAG-tagged GAL4 fusion proteins along with indicated HA-tagged constructs was performed as in *Figure 4C*. (E) Putative complex recruitment mediated by ELL-AF4 interaction. (F) Putative complex recruitment mediated by ELL-EAF1 interaction. (G) Structural requirement of MTM-AF4 pSER fusion and MTMT-EAF1 fusion. Various MTMh/MTMTh constructs fused with EAF1 domains were examined for the transformation of myeloid progenitors as in *Figure 1A*. The online version of this article includes the following figure supplement(s) for figure 5:

**Figure supplement 1.** Roles for AF4/ENL/P-TEFb (AEP) in MLL-ELL-mediated leukemic transformation, related to *Figure 5*.

failed to transform HPCs (*Figure 5G*, see MTMh-EAF1-C). However, an MTM construct fused to the entire EAF1 demonstrated partial transforming properties, which maintained the expression of *Hoxa9* in the early passages, but failed to immortalize HPCs (*Figure 5G*, see MTMh-EAF1). MTMT constructs fused to the EHD fully transformed HPCs (*Figure 5G*, see MTMTh-EAF1 and MTMTh-EAF1-N). This trend is reminiscent of MLL-AF6 which transforms HPCs by homodimerization (*Figure 1B*; *Liedtke et al., 2010*) and suggests that MLL-EAF1 transforms by EHD-mediated homo-dimerization rather than by ELL-EAF1 interaction. Knockdown of *Mllt1* (the gene encoding ENL) in MLL-ELL-ICs perturbed colony formation and *Hoxa9* expression, while its effects on MLL-ENL-ICs were relatively limited presumably because MLL-ENL can directly recruit AEP (*Figure 5—figure supplement 1B*). With these results, we speculated that MLL-ELL transforms HPCs through the recruit-ment of AEP, while MLL-EAF1 transforms via homodimerization in a mechanism similar to MLL-AF6.

## MLL-ELL transforms HPCs via association with AEP, but not with EAF1 or p53

To further evaluate the significance of ELL-AF4 and ELL-EAF1 interactions in MLL-ELL-mediated leu-kemic transformation, we introduced S600A/K606T double mutation (hereafter denoted as SA/KT) to the ELL portion, which was initially predicted to impair ELL-AF4 interaction based on the structural data for the ELL2/AF5Q31 complex (*Qi et al., 2017*). Indeed, the mutation severely attenuated ELL-AF4 interaction (*Figure 6A*, see fGAL4-ELL″ SA/KT). However, a substantial amount of AF4 (mostly the processed forms of AF4) remained associated with the SA/KT mutant, while co-precipitation of EAF1 was completely abolished. These results suggest that SA/KT mutation abolishes the primary contact of ELL for AF4 and EAF1, while there is an additional contact for AF4 that is unaffected by this mutation. Western blotting of the input samples of the chromatin fraction showed that the SA/KT mutant increased the amount of processed forms of AF4 in the chromatin fraction like the intact form (*Figure 6A*, see fGAL4-ELL″ and fGAL4-ELL″SA/KT in the Xpress blot), thus further confirming the interaction with fGAL4-ELL″ SA/KT and AF4. ChIP-qPCR analysis confirmed that exogenously expressed AF4 was recruited to the target chromatin by fGAL4-ELL″ SA/KT, while EAF1 was not (*Figure 6B*). Moreover, fGAL4-ELL″ SA/KT co-precipitated endogenous AF4 and AF5Q31 (*Figure 6—figure supplement 1A*), while it failed to pull down p53, another ELL associating factor (*Wiederschain et al., 2003*). fGAL4-ELL″ SA/KT recruited a substantial amount of endogenous ENL and CyclinT1 to the GAL4-responsive promoter (*Figure 6—figure supplement 1B*), indicating that the SA/KT mutant is competent for loading AEP onto chromatin.

Next, we examined the effects of the SA/KT mutation on the transforming properties of MLL-ELL. Both *Hoxa9* expression and colony-forming potentials were maintained by MLL-ELL SA/KT-ICs (*Figure 6C* and *Figure 6—figure supplement 1C*). Remarkably, the MTM construct fused to ELL car-rying the SA/KT mutation activated *Hoxa9* and immortalized HPCs albeit with low clonogenicity despite the lack of THD2 (*Figure 6C*, see MTMh-ELL″ SA/KT), suggesting that the SA/KT mutation partially compensates for the lack of interaction with the HBO1 complex. MLL-ELL SA/KT induced leukemia in vivo (*Figure 6D*), indicating that direct recruitment of EAF1 or p53 is dispensable and potentially suppressive for MLL-ELL-mediated leukemic transformation. sgRNA competition assays showed that loss of *Eaf1* has a marginal inhibitory effect on proliferation of MLL-ELL-ICs ex vivo, while loss of *Trp53* accelerated it, suggestive of non-essential roles for EAF1 and an inhibitory role for p53 in MLL-ELL-mediated transformation (*Figure 6—figure supplement 1D*). IP-WB analysis of

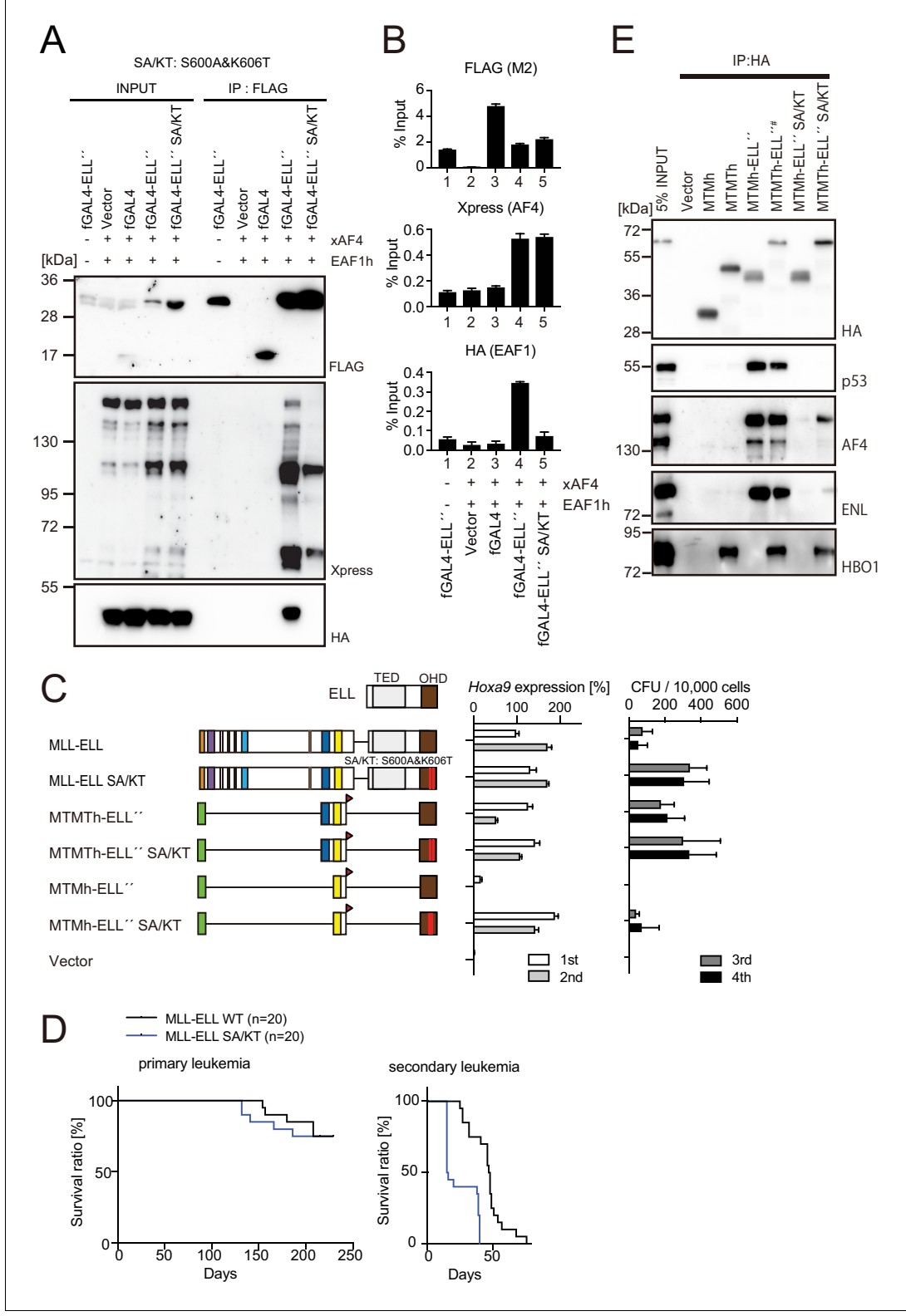

**Figure 6.** MLL-ELL transforms hematopoietic progenitors via association with AF4/ENL/P-TEFb (AEP), but not with EAF1 or p53. (**A**) Mutations of ELL selectively abrogate interaction with EAF1. Immunoprecipitation (IP)-western blotting of the chromatin fraction of HEK293TL cells transiently expressing FLAG-tagged GAL4-ELL proteins with or without S600A/K606T substitutions (SA/KT) along with xAF4 and EAF1h was performed as in *Figure 4D*. (**B**) Recruitment of exogenously expressed AF4 or EAF1 by ELL mutant proteins. Chromatin immunoprecipitation

*Figure 6 continued on next page*

*Figure 6 continued*

(ChIP)-qPCR analysis of HEK293TL cells transiently expressing FLAG-tagged GAL4-ELL proteins with or without the SA/KT mutation along with xAF4 and EAF1h was performed as in *Figure 4C*. (**C**) Effects of the SA/KT mutation on MLL-ELL-mediated leukemic transformation ex vivo. Various MLL-ELL constructs with or without the SA/KT mutation were examined for the transformation of myeloid progenitors as in *Figure 1A*. (**D**) Effects of the SA/KT mutation on MLL-ELL-mediated leukemic transformation in vivo. MLL-ELL or its SA/KT mutant was transduced to c-Kit-positive hematopoietic progenitors and transplanted into syngeneic mice (n = 20). Primary leukemia cells were harvested from the bone marrow and transplanted into secondary recipient mice (n = 20). (**E**) Enhanced interaction of ELL with AEP mediated by the trithorax homology domain 2 (THD2) domain. IP-western blotting of the chromatin fraction of virus-packaging cells, transiently expressing various HA-tagged MTMT-ELL fusion constructs (with or without the SA/KT mutation), was performed. Endogenous proteins co-purified with MLL-ELL proteins were visualized by specific antibodies.

The online version of this article includes the following figure supplement(s) for figure 6:

**Figure supplement 1.** Roles for EAF1 and p53 in MLL-ELL-mediated transformation 6.

the MTMh- or MTMTh-ELL" SA/KT showed that association with AF4 and ENL through the presumed secondary contact is enhanced by the presence of THD2 while p53 association remained abolished (*Figure 6E*). Leukemia cells of MLL-ELL were particularly sensitive to WM1119, a pan MYST family HAT inhibitor (*MacPherson et al., 2020*), compared to other MLL fusion-leukemia cells (*Figure 6—figure supplement 1E*), suggesting that HBO1-mediated protein acetylation may enhance the ELL-AF4 association. In summary, these results suggested that MLL-ELL transforms HPCs via interaction with AF4 family proteins, which is promoted by the HBO1 complex but hindered by other ELL-associated factors.

## NUP98-HBO1 fusion transforms myeloid progenitors through interaction with MLL

NUP98-HBO1 fusion has been found in chronic myelomonocytic leukemia and was shown to induce clinically relevant leukemia in mice (*Hayashi et al., 2019*). NUP98-HBO1 transformed HPCs ex vivo, which was accompanied by high *Hoxa9* expression (*Figure 7A*). Fusion of NUP98 and homeodomain proteins (e.g., NUP98-HOXA9) has been shown to induce leukemia in mouse models (*Kroon et al., 2001*). The homeodomain of HOXA9 possesses a sequence-specific DNA-binding ability. ChIP-qPCR analysis demonstrated that FLAG-tagged NUP98-HOXA9 (fNUP98-HOXA9) bound to the HOXA9-binding site within the HOXA7 locus, whereas fNUP98-HBO1 did not (*Figure 7—figure supplement 1A*), suggesting that HBO1 provides an alternative chromatin targeting function that differs from that of the HOXA9 homeodomain. Because the HBO1 complex binds MLL (*Figure 7B*), we hypothesized that the HBO1 portion might confer a targeting ability through association with MLL. To test this hypothesis, we generated an artificial NUP98 fusion construct in which MENIN is fused to NUP98 and examined its transforming properties. Indeed, NUP98-MENIN transformed HPCs (*Figure 7C*). Artificial NUP98 constructs fused with an HBO complex component (i.e., fNUP98-MEAF6 or -ING5) also transformed HPCs. IP-western blotting showed that these artificial NUP98 fusions were associated with MLL in the chromatin fraction, supporting the hypothesis that NUP98-HBO1 transforms HPCs via interaction with MLL (*Figure 7D*). The NUP98 portion did not co-precipitate MLL in this setting (*Figure 7D*, see fNUP98ΔFP), consistent with a previous report (*Shima et al., 2017*). However, it should be noted that the NUP98 portion was shown to localize in proximity with MLL (*Xu et al., 2016*) and promote the physical interaction of NUP98-HOXA9 with MLL (*Shima et al., 2017*). The sgRNA competition assay demonstrated fNUP98-HBO1-ICs depended on MLL for continuous proliferation (*Figure 7E*), similarly to fNUP98-HOXA9-ICs as reported previously (*Shima et al., 2017*; *Xu et al., 2016*). However, both fNUP98-HOXA9- and fNUP98-HBO1-ICs were mildly sensitive to MENIN-MLL interaction inhibitor (MI-2–2) (*Shi et al., 2012*), and were not rendered completely differentiated unlike MLL-AF10-ICs at the concentration of 10 μM (*Figure 7F*), suggesting that a MENIN-less MLL complex may sufficiently recruit NUP98-HBO1 to the target chromatin. Interestingly, E508Q amino acid substitution, which kills its HAT activity (*Foy et al., 2008*) but retains the binding capacities to MLL (*Figure 7D*), did not impair the transforming property of NUP98-HBO1 (*Figure 7C*). Moreover, WM1119 did not induce complete differentiation of NUP98-HBO1-ICs, as seen in MLL-AF10-ICs, at 10 μM (*Figure 7—figure supplement 1B*). These results

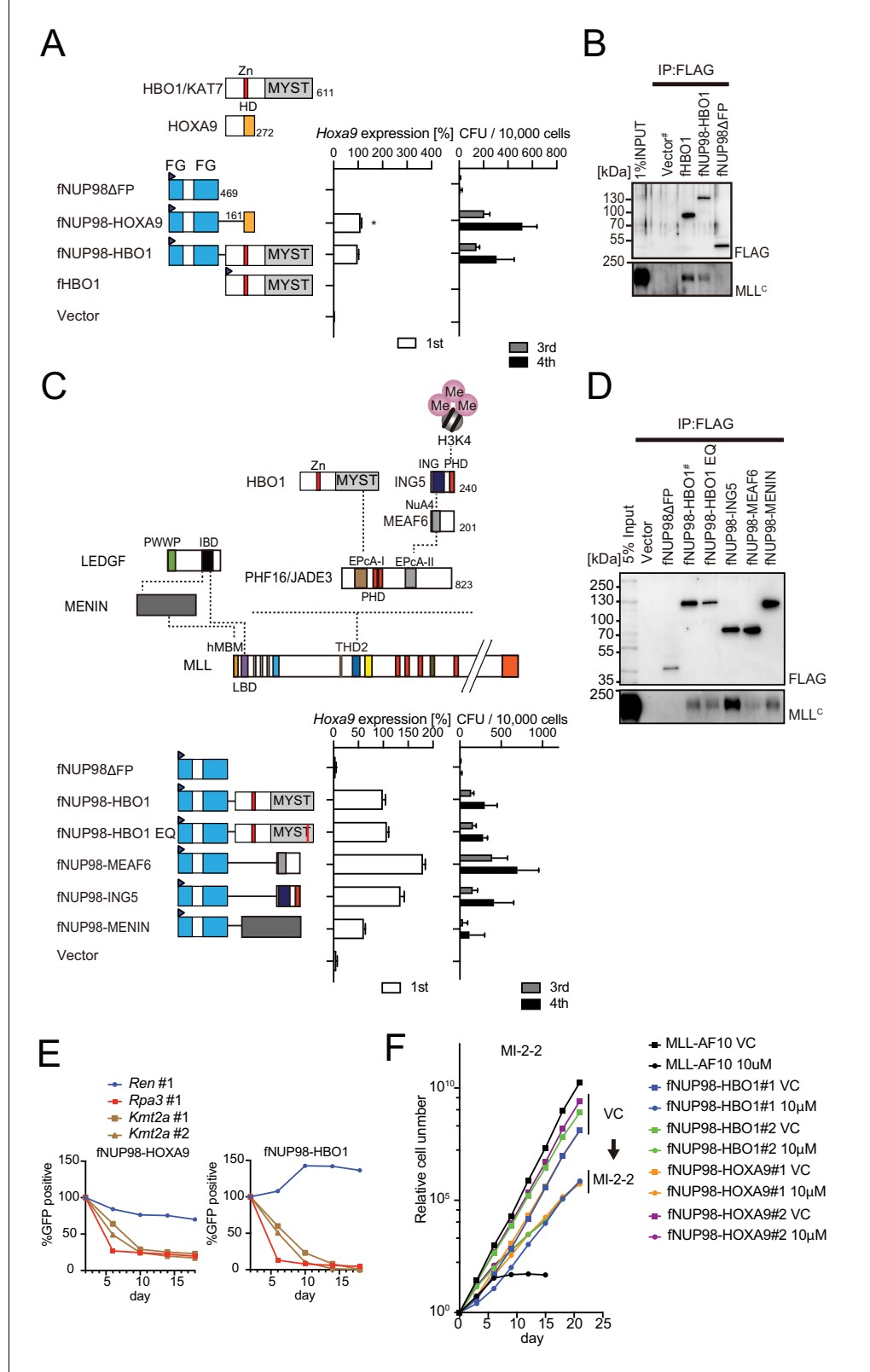

**Figure 7.** Nucleoporin-98 (NUP98)-HBO1 fusion transforms myeloid progenitors through interaction with MLL. (**A**) Structural requirement of NUP98-HBO1 fusion. Various NUP98 fusion constructs were examined for the transformation of myeloid progenitors (along with an HBO1 construct) as in *Figure 1A*. Relative *Hoxa9* expression in first round colonies was analyzed. Asterisk indicates that the qPCR probe detected a human HOXA9 coding

*Figure 7 continued on next page*

*Figure 7 continued*

sequence included in the NUP98-HOXA9 construct in addition to endogenous murine *Hoxa9*. Zn: zinc finger; MYST: MYST HAT domain; HD: homeodomain; FG: phenylalanine-glycine repeat. (**B**) Association with MLL by NUP98-HBO1. The chromatin fraction of virus-packaging cells transiently expressing the indicated transgenes was subjected to immunoprecipitation (IP)-western blotting. Endogenous MLL[C] fragment was detected by a specific anti-MLL antibody. (**C**) Structural requirement of various NUP98 fusions for leukemic transformation. Indicated NUP98 fusion constructs were examined for the transformation of myeloid progenitors as in *Figure 1A*. ING: inhibitor of growth domain of ING4 and ING5; NuA4: histone acetyltransferase subunit NuA4; EPcA-I/II: enhancer of polycomb A domains I and II. (**D**) Association with MLL by various NUP98 fusions. IP-western blotting was performed as in (**B**). (**E**) Requirement of MLL for NUP98 fusion-immortalized cells. sgRNA competition assays for *Kmt2a* were performed on NUP98-HOXA9- and NUP98-HBO1-immortalized cells as in *Figure 2F*. (**F**) Effects of pharmacologic inhibition of MENIN-MLL interaction on NUP98 fusion-immortalized cells. NUP98-HBO1- and NUP98-HOXA9-immortalized cells were cultured in the presence of 10 µM of MI-2–2 MENIN-MLL interaction inhibitor, and their proliferation was monitored every 3 days. MLL-AF10-immortalized cells were also analyzed as a positive control. VC: vehicle control.

The online version of this article includes the following figure supplement(s) for figure 7:

**Figure supplement 1.** Functional differences between nucleoporin-98 (NUP98)-HOXA9 and NUP98-HBO1.

indicate that the interaction with MLL, and not the intrinsic HAT activity, mediates NUP98-HBO1-mediated leukemic transformation.

## Association of MLL with the HBO1 complex is mainly mediated by PHF16 and ING4/5

To dissect the mechanism of association between MLL and the HBO1 complex, we performed IP-WB analyses of the chromatin fractions from HEK293T cells transiently expressing the MLL bait protein (fMLL 869–1124) in combination with each component of the HBO1 complex. Only PHF16 and ING5 were weakly detected in the co-precipitates of the MLL bait protein; however, a very strong association was observed when all of the HBO1 complex components were co-expressed (*Figure 8A*), suggesting that the association of MLL with the HBO1 complex is mediated via multiple contacts likely provided by PHF16 and ING5. IP-WB analysis wherein each component of the HBO1 complex was omitted indicated the PHF16 and ING5 are indeed the most influential components for the MLL-HBO1 complex association (*Figure 8B*). Notably, the omission of HBO1 reduced the association albeit to a much lesser extent, which can be rescued by adding back the C-terminal half of HBO1 that mediates interaction with PHF16 and ING5. The omission of MEAF6 also reduced the association substantially, indicating that the formation of the entire HBO1 complex maximizes the potential of the MLL-HBO1 complex association. Deletion of the ING domain from ING5 resulted in the loss of the MLL-HBO1 complex association, suggesting that the ING domain may provide a direct contact for MLL interaction. Point mutations of the aromatic cage of the PHD finger of ING5 (Y188A and W211A) that directly binds to H3K4me3 marks (*Champagne et al., 2008*) resulted in the loss of the MLL-HBO1 complex association, which could be rescued by ING4 (*Figure 8C*), indicating that ING4/5-mediated chromatin binding to H3K4me3 marks is required for efficient MLL-HBO1 complex association. These results suggest that ING4/5-mediated chromatin binding precedes the MLL-HBO1 complex association, and the MLL and HBO1 complexes stably associate with each other on the chromatin containing H3K4me3 marks to promote subsequent AEP/SL1-mediated transcriptional activation by MLL fusions (*Figure 8D*).

## Discussion

Most MLL fusion proteins constitutively recruit AEP by a variety of mechanisms to immortalize HPCs (*Takahashi and Yokoyama, 2020*). However, it was unclear how MLL-ELL exerts its oncogenic properties, as it was reported that an artificial MLL-EAF1 fusion immortalized HPCs (*Luo et al., 2001*). AF4 and EAF family proteins share structural similarities, both of which have the SDE motif and are shown to bind to SL1 (*Figure 5A–C*; *Okuda et al., 2016*). We previously demonstrated that the combination of SDE and NKW motifs is required for transcriptional activation (*Okuda et al., 2015*). Because EAF1 does not have an NKW motif, EAF1 is incompetent for transcriptional activation, and

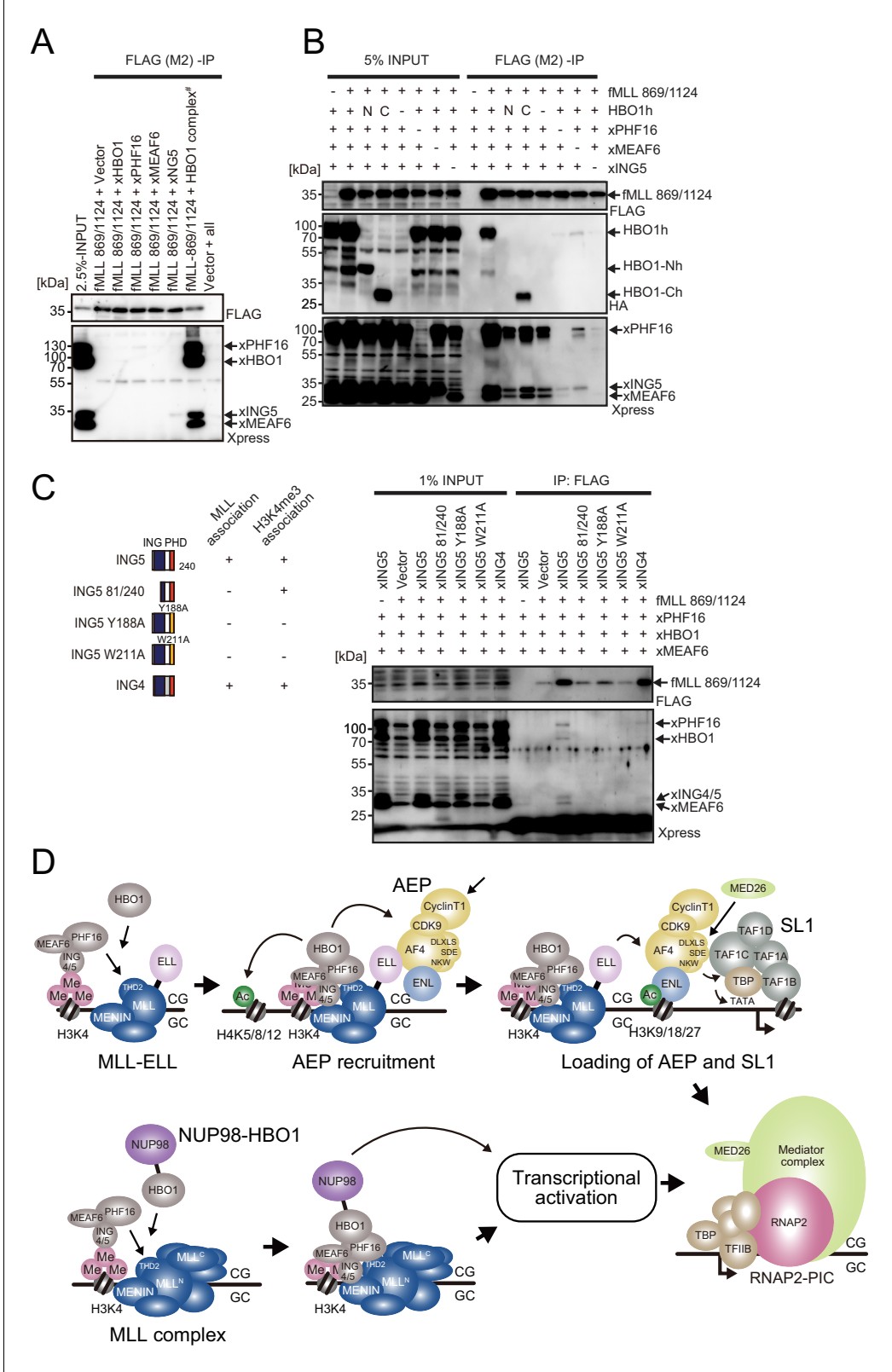

**Figure 8.** MLL-HBO1 complex association is mediated by multiple contacts and in a chromatin-bound context containing H3K4 methylation. (**A**) Association of the MLL bait protein with the HBO1 complex. Immunoprecipitation (IP)-western blotting of the chromatin fraction of HEK293T cells transiently expressing FLAG-tagged MLL 869/1124 construct and Xpress-tagged HBO1 complex components was performed. (**B**) Influence of each

*Figure 8 continued on next page*

Figure 8 continued

HBO1 complex component for MLL-HBO1 complex association. (C) Domain mapping of ING5 responsible for MLL-HBO1 complex association. (D) Models of MLL-ELL- and nucleoporin-98 (NUP98)-HBO1-mediated gene activation.

therefore MTMh-EAF1-C did not transform HPCs (*Figure 5G*). Instead, EAF1 homodimerizes through its EHD. The MTMT-EHD fusion (e.g., MTMTh-EAF1-N) transformed HPCs in a THD2-dependent manner, similar to MLL-AF6, which is known to transform HPCs via homodimerization (*Liedtke et al., 2010*). Thus, we concluded that MLL-EAF1 transforms HPCs via homodimerization. The S600A/K606T double mutation on ELL completely abrogated the interaction with EAF1, while retaining some binding capacity to AF4 and did not impair MLL-ELL-mediated transformation. Thus, we concluded that MLL-ELL transforms HPCs via association with AF4 family proteins, but not with EAF1 (*Figure 8D*).

ChIP-qPCR analysis indicated that ELL recruited AF4 and ENL to the target chromatin but was unable to further recruit SL1 and MED26 (*Figure 5D*), suggesting that the AEP complex must dissociate from ELL to function as a transcriptional activator (*Figure 5E*). Thus, MLL-ELL is a loading factor of the AEP complex, but not a nucleating factor (like MLL-ENL). In our previous study, we showed that MLL-AF10 is also an AEP loading factor that loads ENL onto chromatin (*Okuda et al., 2017*). It remains unclear how MLL-AF6, a dimer type of MLL fusion, affects the function of AEP and transforms HPCs. Nonetheless, ChIP-qPCR analysis showed that MLL-AF6 colocalized with AEP components at its target promoters (*Yokoyama et al., 2010*), suggesting that MLL-AF6 also loads AEP to the target chromatin via unknown mechanisms. All of the presumed AEP loading factor type MLL fusions (i.e., MLL-ELL, MLL-AF10, and MLL-AF6) showed susceptibility to *Mllt1* knockdown, whereas the AEP nucleating factor type MLL fusions (e.g., MLL-ENL) were relatively resistant (*Figure 5—figure supplement 1B*; *Okuda et al., 2017*; *Yokoyama et al., 2010*). Moreover, the AEP loading factor type MLL fusions exhibited relatively strong dependency on THD2, which mediates HBO1 complex recruitment (*Figure 1*). These results suggest that the HBO1 complex promotes loading of the AEP complex onto target chromatin (*Figure 8D*). Hence, the inhibitors that interfere with the MLL-HBO complex interaction may be highly effective for the AEP loading factor type MLL fusions.

Like MLL, NUP98 fuses with a variety of partners (*Gough et al., 2011*). NUP98 fusion partners can be roughly subdivided into three groups: the homeodomain-containing protein type (e.g., NUP98-HOXA9), the chromatin reader type (e.g., NUP98-HBO1, -LEDGF, -MLL, -NSD1, -PHF23, -BPTF, and -KDM5A), and the coiled-coil structure containing protein type. Among the chromatin reader type NUP98 fusions, MLL and LEDGF are components of the MLL complex (*Yokoyama and Cleary, 2008*), which targets previously transcribed CpG-rich promoters through association with unmethylated CpGs, di/tri-methylated histone H3 lysine 36 (H3K36me2/3) (*Okuda et al., 2014*), and di/tri-methylated histone H3 lysine 4 (H3K4me2/3) recognized by the PHD finger 3 of MLL (*Milne et al., 2010*; *Wang et al., 2010*). NUP98-NSD1 was shown to activate HOX-A genes through the association with methylated H3K36 marks (*Wang et al., 2007*), and NUP98-PHF23, -BPTF, and-KDM5A were through association with H3K4me2/3 marks (*Zhang et al., 2020*). Thus, the chromatin reader type NUP98 fusions confer a targeting ability to the chromatin of the HOX-A loci by interacting with specific chromatin modifications or chromatin-binding proteins. A subset of NUP98 fusions including NUP98-HBO1 and -LEDGF appear to target through interaction with MLL at least in part. This was supported by the immortalization of HPCs by the artificial fusion of NUP98 and MENIN (*Figure 7C*). It has been reported that a portion of MLL target genes is regulated in a MENIN-independent manner (*Artinger et al., 2013*) and an MLL mutant protein lacking the MENIN-binding motif but retaining the CXXC domain and the PHD finger 3 localized at the *HOXA9* locus presumably through the association with unmethylated CpGs and H3K4me2/3 (*Milne et al., 2010*; *Wang et al., 2010*), suggesting that MLL can bind to some of its target chromatin in a MENIN-independent manner. The NUP98-MLL fusion protein expressed in leukemia patients does not retain the structures responsible for interaction with MENIN and LEDGF (*Kaltenbach et al., 2010*), suggesting that the MLL portion in NUP98-MLL confers its targeting ability in a MENIN-independent manner. The proliferation of NUP98-HBO1-ICs was dependent on MLL (*Figure 7E*) and was slowed but not entirely blocked by the MENIN-MLL interaction inhibitor (*Figure 7F*). These results indicate that a MENIN-less MLL complex may sufficiently recruit NUP98-HBO1 possibly through the association with H3K4me3 marks via the PHD finger 3 of MLL. The presence of H3K4me3 marks at the target

promoter is also a prerequisite for the HBO1 complex-MLL association on chromatin as they are recognized by the PHD finger of ING4/5 subunits (*Figure 3F* and *Figure 8C*; *Champagne et al., 2008*). Taken together, these results suggest that a subset of NUP98 fusions target the HOX-A loci via association with MLL and H3K4me3 marks and activate transcription by NUP98-mediated functions (*Figure 8D*).

This study demonstrated that various MLL fusions and NUP98 fusions transform HPCs via HBO1-MLL interaction (*Figure 8D*). Hence, we propose that the binding modules in this interaction can be a good molecular target for drug development. HBO1-MLL interaction inhibitors would complement the therapeutic effects of the emerging MENIN-MLL interaction inhibitors (*Klossowski et al., 2019*; *Krivtsov et al., 2019*) to treat malignant leukemia with subsets of MLL and NUP98 fusions.

# Materials and methods

## Materials availability
Materials generated in this study will be provided upon request.

## Experimental model and subject details
### Vector constructs
For protein expression vectors, cDNAs obtained from Kazusa Genome Technologies Inc (*Nagase et al., 2008*) were modified by PCR-mediated mutagenesis and cloned into the pMSCV vector (for virus production) or pCMV5 vector (for transient expression) by restriction enzyme digestion and DNA ligation. The MSCV-neo MLL-ENL and MLL-AF10 vectors have been previously described (*Okuda et al., 2017*). sgRNA expression vectors were constructed using the pLKO5. sgRNA.EFS.GFP vector (*Heckl et al., 2014*). shRNA expression vectors were constructed using a pLKO.1 vector or were purchased from Dharmacon. The target sequences of sgRNA are listed in *Supplementary file 1a and b*.

### Cell lines
HEK293T cells were a gift from Michael Cleary and were authenticated by the JCRB Cell Bank in 2019. HEK293TN cells were purchased from System Biosciences. The cells were cultured in Dulbecco's modified Eagle's medium (DMEM), supplemented with 10% fetal bovine serum (FBS) and penicillin-streptomycin (PS). The Platinum-E (PLAT-E) ecotropic virus-packaging cell line—a gift from Toshio Kitamura (*Morita et al., 2000*)—was cultured in DMEM supplemented with 10% FBS, puromycin, blasticidin, and PS. The human leukemia cell lines HB1119 and REH, gifts from Michael Cleary (*Tkachuk et al., 1992*), were cultured in RPMI 1640 medium supplemented with 10% FBS and PS. HEK293T–LUC cells were generated by transduction of the lentivirus carrying pLKO1-puro-FR-LUC, as described previously (*Okuda et al., 2015*). Murine myeloid progenitors immortalized by various transgenes were cultured in RPMI 1640 medium supplemented with 10% FBS, and PS containing murine stem cell factors (mSCFs), murine interleukin-3 (mIL-3), and murine granulocyte-macrophage colony-stimulating factor (mGM-CSF; 1 ng/mL of each). Cells were cultured in the incubator at 37°C and 5% $CO_2$ and routinely tested for mycoplasma using the MycoAlert Mycoplasma detection kit (Lonza).

### Western blotting
Western blotting was performed as previously described (*Yokoyama et al., 2002*). Briefly, proteins were separated electrophoretically in an acrylamide gel and were transblotted onto nitrocellulose sheets using a mini transblot cell (Bio-Rad). The nitrocellulose sheets were blocked with 5% skimmed milk in T-PBS (phosphate-buffered saline containing 0.1% Tween 20) for 1 hr, rinsed twice with T-PBS, and incubated with primary antibodies suspended in 5% skim milk in T-PBS overnight. The blots were then washed twice with T-PBS and incubated with peroxidase-conjugated secondary antibodies for 2 hr. Chemiluminescence was performed using the ECL chemiluminescence reagent (GE Healthcare). The antibodies used in this study are listed in Key resources table.

## Virus production

Ecotropic retrovirus was produced using PLAT-E packaging cells (*Morita et al., 2000*). Lentiviruses were produced in HEK293TN cells using the pMDLg/pRRE, pRSV-rev, and pMD2.G vectors, all of which were gifts from Didier Trono (*Dull et al., 1998*). The virus-containing medium was harvested 24–48 hr following transfection and used for viral transduction.

## Myeloid progenitor transformation assay

The myeloid progenitor transformation assay was carried out as previously described (*Lavau et al., 1997*; *Okuda and Yokoyama, 2017b*). Bone marrow cells were harvested from the femurs and tibiae of 5-week-old female C57BL/6J mice. c-Kit-positive cells were enriched using magnetic beads conjugated with an anti-c-Kit antibody (Miltenyi Biotec), transduced with a recombinant retrovirus by spinoculation, and then plated ($4 \times 10^4$ cells/sample) in a methylcellulose medium (Iscove's modified Dulbecco's medium, 20% FBS, 1.6% methylcellulose, and 100 µM β-mercaptoethanol) containing mSCF, mIL-3, and mGM-CSF (10 ng/mL each). During the first culture passage, G418 (1 mg/mL) or puromycin (1 µg/mL) was added to the culture medium to select the transduced cells. *Hoxa9* expression was quantified by qRT-PCR after the first passage. Cells were then re-plated once every 5 days with fresh medium. Colony-forming units were quantified per $10^4$ plated cells at each passage.

## In vivo leukemogenesis assay

In vivo leukemogenesis assay was carried out as previously described (*Lavau et al., 1997*; *Okuda and Yokoyama, 2017a*). Cells positive for c-Kit ($2 \times 10^5$), prepared from mouse femurs and tibiae, were transduced with retrovirus by spinoculation and intravenously transplanted into sublethally irradiated (5–6 Gy) 8-week-old female C57BL/6JJcl (C57BL/6J) mice. Moribund mice were euthanized, and the cells isolated from their bone marrow were cultured in methylcellulose medium used for myeloid progenitor transformation assays for more than three passages to remove untransformed cells and then subjected to secondary transplantation. For secondary leukemia, leukemia cells ($2 \times 10^5$) cultured ex vivo were transplanted in the same manner as the primary transplantation. This protocol was approved by the National Cancer Center Institutional Animal Care and Use Committee.

## qRT-PCR

Total RNA was isolated using the RNeasy Mini Kit (Qiagen) and reverse-transcribed using the Superscript III First Strand cDNA Synthesis System (Thermo Fisher Scientific) with oligo (dT) primers. Gene expression was analyzed by qPCR using TaqMan probes (Thermo Fisher Scientific). Relative expression levels were normalized to those of *GAPDH/Gapdh* and determined using a standard curve and the relative quantification method, according to the manufacturer's instructions (Thermo Fisher Scientific). Commercially available PCR probe sets used in this study are listed in Key resources table.

## Fractionation-assisted chromatin immunoprecipitation

Chromatin fractions from HEK293T and HB1119 cells were prepared using the fanChIP method, as described previously (*Miyamoto and Yokoyama, 2021*; *Okuda et al., 2014*). Cells were suspended in CSK buffer (100 mM NaCl, 10 mM PIPES [pH 6.8], 3 mM MgCl$_2$, 1 mM EGTA, 0.3 M sucrose, 0.5% Triton X-100, 5 mM sodium butyrate, 0.5 mM DTT, and protease inhibitor cocktail) and centrifuged ($400 \times$ *g* for 5 min, at 4°C) to remove the soluble fraction. The pellet was resuspended in MNase buffer (50 mM Tris-HCl [pH 7.5], 4 mM MgCl$_2$, 1 mM CaCl$_2$, 0.3 M sucrose, 5 mM sodium butyrate, 0.5 mM DTT, and protease inhibitor cocktail) and treated with MNase at 37°C for 3–6 min to obtain oligonucleosomes. MNase reaction was then stopped by adding EDTA (pH 8.0) to a final concentration of 20 mM. An equal amount of lysis buffer (250 mM NaCl, 20 mM sodium phosphate [pH 7.0], 30 mM sodium pyrophosphate, 5 mM EDTA, 10 mM NaF, 0.1% NP-40, 10% glycerol, 1 mM DTT, and EDTA-free protease inhibitor cocktail) was added to increase solubility. The chromatin fraction was cleared by centrifugation (15,000 rpm for 5 min, 4°C) and subjected to immunoprecipitation with specific antibodies and Protein-G magnetic microbeads (Invitrogen) or with anti-FLAG M2 antibody-conjugated beads (Key resource table). Immunoprecipitates were then washed five times with washing buffer (1:1 mixture of lysis buffer and MNase buffer with 20 mM EDTA) and eluted in elution buffer (1% SDS and 50 mM NaHCO$_3$). The eluted material was analyzed by multiple methods

including western blotting, qPCR, and deep sequencing. Optionally, the immunoprecipitates were washed twice with washing buffer, twice with MNase buffer, and then treated with DNase I (Qiagen) for 15 min at room temperature. The immunoprecipitates were further washed four times with washing buffer and eluted using elution buffer. The eluted materials were subjected to western blotting and SYBR green staining.

## ChIP-qPCR and ChIP-seq

The eluted material obtained by fanChIP was extracted using phenol/chloroform/isoamyl alcohol. DNA was precipitated with glycogen, dissolved in TE buffer, and analyzed by qPCR (ChIP-qPCR) or deep sequencing (ChIP-seq). The qPCR probe/primer sequences are listed in *Supplementary file 1c*. For deep sequencing, DNA was further fragmented (~150 bp) using the Covaris M220 DNA shearing system (M and M Instruments Inc). Deep sequencing was then performed using the TruSeq ChIP Sample Prep Kit (Illumina) and HiSeq2500 (Illumina) at the core facility of Hiroshima University. Data were visualized using Integrative Genome Viewer (The Broad Institute). Raw reads in FASTQ format were trimmed using Cutadapt and aligned to the reference genome hg19 with BWA (*Li and Durbin, 2009*; *Martin, 2011*). Accession numbers and sample IDs are listed in *Supplementary file 1d*.

## Liquid chromatography-tandem mass spectrometry analysis

Proteins were digested with trypsin and tandem mass spectrometry was performed using an LTQ Orbitrap ELITE ETD mass spectrometer (Thermo Fisher Scientific), as described previously (*Okuda et al., 2017*).

## sgRNA competition assay

*Cas9* was introduced via lentiviral transduction using the pKLV2-EF1a-Cas9Bsd-W vector (*Tzelepis et al., 2016*). Cas9-expressing stable lines were established with blasticidin (10–30 µg/mL) selection. The sequences of sgRNAs are listed in *Supplementary file 1a and b*. The targeting sgRNA was co-expressed with GFP via lentiviral transduction using pLKO5.sgRNA.EFS.GFP vector (*Heckl et al., 2014*). Percentages of GFP-positive cells were initially determined by FACS analysis at 2 days after sgRNA transduction and then measured every 4 days.

## Accession numbers

Deep sequencing data used in this study have been deposited in the DNA Data Bank of Japan (DDBJ) Sequence Read Archive under the accession numbers listed in *Supplementary file 1d* (ChIP-seq, CIRA-seq, and RNA-seq). The mass spectrometry data were deposited to Japan ProteOme STandard Repository (jPOSTrepo) under the accession number JPST001262 (PXID:PXD027400).

## Statistics

Statistical analysis was performed using GraphPad Prism7 software. Data are presented as the mean with standard deviation. Multiple comparisons were performed by two-way ANOVA. The statistical tests were two-sided. Mice transplantation experiments were analyzed by the log-rank test and Bonferroni correction was applied for multiple comparisons. p-values < 0.05 were considered statistically significant. n.s.: $p > 0.05$, *$p \leq 0.05$, **$p \leq 0.01$, ***$p \leq 0.001$, and ****$p \leq 0.0001$.

## Study approval

All animal experimental protocols were approved by the National Cancer Center (Tokyo, Japan) Institutional Animal Care and Use Committee (T17501).

## Acknowledgements

We thank Ikuko Yokoyama, Hagumu Sato, Kanae Ito, Makiko Okuda, Yuzo Sato, Megumi Nakamura, Etsuko Kanai, Aya Nakayama, and Ayako Yokoyama for technical assistance. We also thank the Shonai Regional Industry Promotion Center members for their administrative support. This work was supported by the Japan Society for the Promotion of Science (JSPS) KAKENHI grants (16H05337, 19H03694 to AY). This work was also supported in part by research funds from the Yamagata

prefectural government, the City of Tsuruoka, Dainippon Sumitomo Pharma Co. Ltd., and the Friends of Leukemia Research Fund.

## Additional information

### Competing interests

Akihiko Yokoyama: A.Y. received a research grant from Dainippon Sumitomo Pharma Co. Ltd. The other authors declare that no competing interests exist.

### Funding

| Funder | Grant reference number | Author |
| --- | --- | --- |
| Japan Society for the Promotion of Science | 16H05337 | Akihiko Yokoyama |
| Japan Society for the Promotion of Science | 19H03694 | Akihiko Yokoyama |
| Yamagata Health Support Association | | Akihiko Yokoyama |
| the City of Tsuruoka | | Akihiko Yokoyama |
| Leukemia Research Foundation | | Akihiko Yokoyama |
| Dainippon Sumitomo Pharma Co., Ltd. | | Akihiko Yokoyama |

The funders had no role in study design, data collection and interpretation, or the decision to submit the work for publication.

### Author contributions

Satoshi Takahashi, Conceptualization, Resources, Formal analysis, Investigation, Visualization, Methodology, Writing - original draft, Writing - review and editing; Akinori Kanai, Resources, Data curation, Software, Formal analysis, Visualization, Methodology; Hiroshi Okuda, Resources, Formal analysis, Validation; Ryo Miyamoto, Formal analysis, Investigation; Yosuke Komata, Formal analysis, Investigation, Visualization; Takeshi Kawamura, Data curation, Formal analysis, Visualization, Methodology; Hirotaka Matsui, Resources, Formal analysis, Methodology; Toshiya Inaba, Resources, Methodology; Akifumi Takaori-Kondo, Supervision; Akihiko Yokoyama, Conceptualization, Resources, Data curation, Formal analysis, Supervision, Funding acquisition, Validation, Investigation, Visualization, Methodology, Writing - original draft, Project administration, Writing - review and editing

### Author ORCIDs

Akinori Kanai ⓘD https://orcid.org/0000-0003-1555-6768
Akifumi Takaori-Kondo ⓘD http://orcid.org/0000-0001-7678-4284
Akihiko Yokoyama ⓘD https://orcid.org/0000-0002-5639-8068

### Ethics

Animal experimentation: All animal experimental protocols were approved by the National Cancer Center (Tokyo Japan) Institutional Animal Care and Use Committee.

### Decision letter and Author response

Decision letter https://doi.org/10.7554/eLife.65872.sa1
Author response https://doi.org/10.7554/eLife.65872.sa2

## Additional files

### Supplementary files

- Source data 1. Gel images.
- Supplementary file 1. Sequence data.
- Transparent reporting form

### Data availability

ChIP-seq data, CIRA-seq data, and RNA-seq data have been deposited at the DDBJ (DNA Data Bank of Japan) Sequence Read Archive under the accession numbers (DRA010818, DRA004871, DRA004872, DRA010819, DRA008732, DRA008734, and DRA004874) and sample IDs listed in Supplementary file 1.The mass spectrometry data were deposited to Japan ProteOme STandard Repository (jPOSTrepo) under the accession number JPST001262 (PXID:PXD027400).

The following datasets were generated:

| Author(s) | Year | Dataset title | Dataset URL | Database and Identifier |
|---|---|---|---|---|
| HIROSHIMA | 2020 | Genomic localization of various factors/modifications in HB1119 cells | https://ddbj.nig.ac.jp/public/ddbj_database/dra/fastq/DRA010/DRA010818/ | DDBJ, DRA010818 |
| HIROSHIMA | 2020 | Genomic localization of various FLAG-tagged proteins in HEK293T cells | https://ddbj.nig.ac.jp/public/ddbj_database/dra/fastq/DRA010/DRA010819/ | DDBJ, DRA010819 |
| HIROSHIMA | 2021 | Genomic localization of various proteins in HB1119 cells | https://ddbj.nig.ac.jp/public/ddbj_database/dra/fastq/DRA012/DRA012472/ | DDBJ, DRA012472 |
| HIROSHIMA | 2021 | Genomic localization of various proteins in HEK293T cells | https://ddbj.nig.ac.jp/public/ddbj_database/dra/fastq/DRA012/DRA012473/ | DDBJ, DRA012473 |

The following previously published datasets were used:

| Author(s) | Year | Dataset title | Dataset URL | Database and Identifier |
|---|---|---|---|---|
| HIROSHIMA | 2016 | Genomic localization of various factors/modificaions in HB1119 cells | https://ddbj.nig.ac.jp/public/ddbj_database/dra/fastq/DRA004/DRA004871/ | DDBJ, DRA004871 |
| HIROSHIMA | 2016 | Genomic localization of various factors/modificaions in 293T cells | https://ddbj.nig.ac.jp/public/ddbj_database/dra/fastq/DRA004/DRA004872/ | DDBJ, DRA004872 |
| HIROSHIMA | 2019 | Expression profiles of HB1119 and 293T cell lines | https://ddbj.nig.ac.jp/public/ddbj_database/dra/fastq/DRA004/DRA004874/ | DDBJ, DRA004874 |
| HIROSHIMA | 2019 | ChIP-seq analysis of HEK293T cell lines | https://ddbj.nig.ac.jp/public/ddbj_database/dra/fastq/DRA008/DRA008732/ | DDBJ, DRA008732 |
| HIROSHIMA | 2019 | CIRA-seq analysis of the HEK293T cell line | https://ddbj.nig.ac.jp/public/ddbj_database/dra/fastq/DRA008/DRA008734/ | DDBJ, DRA008734 |

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

# Appendix 1

**Appendix 1—key resources table**

| Reagent type (species) or resource | Designation | Source or reference | Identifiers | Additional information |
|---|---|---|---|---|
| Other | MNase | Sigma-Aldrich | Cat#:N3755-200U | |
| Other | Protein-G Magnetic beads | Thermo Fisher Scientific | Cat#:1004D | |
| Other | FLAG M2 antibody-conjugated beads | Sigma-Aldrich | Cat#:M8823 RRID:AB_2637089 | |
| Other | c-Kit magnetic beads | Miltenyi Biotec | Cat#:130-091-224 RRID:AB_2753213 | |
| Other | MI-2–2 | Calbiochem | Cat#:444825 | |
| Other | WM1119 | Enamine | Cat#:EN300-1719156 | |
| Antibody | MLL(N) | Cell Signaling Technology | Cat#:14689 RRID:AB_2688009 | WB(1000:1) |
| | | | | ChIP(1 µL/400 µL) |
| Antibody | MLL(N) | Generated in-house *Yokoyama et al., 2002* | rpN1 | ChIP(1 µg/400 µL) |
| Antibody | MOZ | Active motif | Cat#:39868 | ChIP(1 µg/400 µL) |
| Antibody | AF4 | Santa Cruz Biotechnology | Cat#:sc-49350 RRID:AB_2226113 | ChIP(1 µg/400 µL) |
| Antibody | TAF1C | Bethyl Laboratories | Cat#:A303-698A RRID:AB_11203194 | ChIP(1 µg/400 µL) |
| Antibody | AF17 | Bethyl Laboratories | Cat#:A302-198A RRID:AB_1659777 | ChIP(1 µg/400 µL) |
| Antibody | DOT1L | Bethyl Laboratories | Cat#:A300-953A RRID:AB_805775 | ChIP(1 µg/400 µL) |
| Antibody | CyclinT1 | Santa Cruz Biotechnology | Cat#:sc-8127 RRID:AB_2073892 | ChIP(1 µg/400 µL) |
| Antibody | ENL | Cell Signaling Technology | Cat#:14893S | WB(1000:1) |
| | | | | ChIP(5 µL/400 µL) |
| Antibody | Histone H3K4me3 | Active motif | Cat#:39159 RRID:AB_2615077 | ChIP(1 µg/400 µL) |
| Antibody | Histone H3K18ac | Abcam | Cat#:ab1191 RRID:AB_298692 | ChIP(1 µg/400 µL) |
| Antibody | RNAP2 non-P | Abcam | 8WG16/Cat#:ab817 RRID:AB_306327 | ChIP(1 µg/400 µL) |
| Antibody | RNAP2 Ser5-P | Millipore | Cat#:05–623 AB_309852 | ChIP(1 µg/400 µL) |
| Antibody | FLAG | Sigma-Aldrich | Cat#:F3165 RRID:AB_259529 | ChIP(1 µg/400 µL) |
| Antibody | FLAG | Sigma-Aldrich | Cat#:F7425 RRID:AB_439687 | WB(1000:1) |
| Antibody | HA | Roche | 3F10/Cat#:11867423001 RRID:AB_390918 | WB(1000:1) |
| | | | | ChIP(0.2 µg/400 µL) |
| Antibody | Xpress | Santa Cruz Biotechnology | Cat#:sc-7270 RRID:AB_675763 | WB(1000:1) |
| | | | | ChIP(1 µg/400 µL) |
| Antibody | ELL | Cell Signaling Technology | Cat#:14468S RRID:AB_2798489 | WB(1000:1) |

*Continued on next page*

*Appendix 1—key resources table continued*

| Reagent type (species) or resource | Designation | Source or reference | Identifiers | Additional information |
|---|---|---|---|---|
| Antibody | HBO1 | Abcam | Cat#:70183 RRID:AB_1269226 | WB(1000:1) |
| | | | | ChIP(1 μg/400 μL) |
| Antibody | PHF16 | Abcam | Cat#:129495 RRID:AB_11157084 | WB(1000:1) |
| | | | | ChIP(1 μg/400 μL) |
| Antibody | MEAF6 | STJ | Cat#:116836 | WB(1000:1) |
| | | | | ChIP(1 μg/400 μL) |
| Antibody | ING4 | Abcam | Cat#:108621 RRID:AB_10860023 | WB(1000:1) |
| | | | | ChIP(1 μg/400 μL) |
| Antibody | MLL(C) | Cell Signaling Technology | Cat#:14197S RRID:AB_2688010 | WB(1000:1) |
| Antibody | AF4 | Bethyl Laboratories | Cat#:A302-344A RRID:AB_1850255 | WB(1000:1) |
| Antibody | AF5Q31 | Bethyl Laboratories | Cat#:A302-538A RRID:AB_1998985 | WB(1000:1) |
| Antibody | P53 | Santa Cruz Biotechnology | Cat#:sc-126 RRID:AB_628082 | WB(1000:1) |
| Sequence-based reagent | *GAPDH* | Thermo Fisher Scientific | Hs02786624_g1 | |
| Sequence-based reagent | *HOXA9* | Thermo Fisher Scientific | Hs00365956_m1 | |
| Sequence-based reagent | *MYC* | Thermo Fisher Scientific | Hs01570247_m1 | |
| Sequence-based reagent | *RPL13A* | Thermo Fisher Scientific | Hs04194366_g1 | |
| Sequence-based reagent | *CDKN2C* | Thermo Fisher Scientific | Hs00176227_m1 | |
| Sequence-based reagent | *Gapdh* | Thermo Fisher Scientific | Mm99999915_g1 | |
| Sequence-based reagent | *Hoxa9* | Thermo Fisher Scientific | Mm00439364_m1 | |
| Sequence-based reagent | *Mllt1* | Thermo Fisher Scientific | Mm00452080_m1 | |
| Sequence-based reagent | sgRNAs | This study | | See *Supplementary file 1* |
| Sequence-based reagent | Custom made qPCR primers | Thermo Fisher Scientific | | See *Supplementary file 1* |
| Recombinant DNA reagent | pMSCV-neo | Clontech | Cat#:634401 | |
| Recombinant DNA reagent | pLKO5.EFS.GFP | Addgene (gift from Benjamin Ebert) *Heckl et al., 2014* | Addgene #57822 RRID:Addgene_57822 | |
| Recombinant DNA reagent | pKLV2-Cas9.bsd | Addgene (gift from Kosuke Yusa) *Tzelepis et al., 2016* | Addgene #68343 RRID:Addgene_68343 | |
| Recombinant DNA reagent | pMDLg/pRRE | Addgene (gift from Didier Trono) *Dull et al., 1998* | Addgene #12251 RRID:Addgene_12251 | |

*Continued on next page*

*Appendix 1—key resources table continued*

| Reagent type (species) or resource | Designation | Source or reference | Identifiers | Additional information |
|---|---|---|---|---|
| Recombinant DNA reagent | pRSV-rev | Addgene (gift from Didier Trono) *Dull et al., 1998* | Addgene #12253 RRID:Addgene_12253 | |
| Recombinant DNA reagent | pMD2.G | Addgene (gift from Didier Trono) *Dull et al., 1998* | Addgene #12259 RRID:Addgene_12259 | |
| Recombinant DNA reagent | pLKO.1-puro | Addgene (gift from Bob Weinberg) *Stewart et al., 2003* | Addgene #8453 RRID:Addgene_84530 | |
| Recombinant DNA reagent | pLKO.1-sh-*Mllt1*-puro | GE Healthcare | TRCN0000084405 | |
| Software, algorithm | GraphPad Prism7 | GraphPad Software Inc | RRID:SCR_002798 | |
| Software, algorithm | Integrative Genomics Viewer | | *Thorvaldsdóttir et al., 2013* | RRID:SCR_011793 |
| Software, algorithm | Cutadapt | *Martin, 2011* | RRID:SCR_011841 | |
| Software, algorithm | BWA | *Li and Durbin, 2009* | RRID:SCR_010910 | |
| Cell line (*Homo sapiens*) | Human: HEK293T | Gift from Michael Cleary | | Authenticated by JCRB Cell Bank in 2019 |
| Cell line (*Homo sapiens*) | Human: HB1119 | Gift from Michael Cleary *Tkachuk et al., 1992* | | The cell line was verified by the expression of MLL-ENL |
| Cell line (*Homo sapiens*) | Human: REH | Gift from Michael Cleary *Yokoyama et al., 2004* | RRID:CVCL_1650 | Authenticated by JCRB Cell Bank in 2021 |
| Cell line (*Homo sapiens*) | Human: PLAT-E | Gift from Toshio Kitamura *Morita et al., 2000* | RRID:CVCL_B488 | |
| Cell line (*Homo sapiens*) | Human: HEK293TN | System Bioscience | Cat#:LV900A-1 RRID:CVCL_UL49 | |
| Cell line (*Homo sapiens*) | Human: HEK293T dMLLc3 | Generated in-house *Miyamoto et al., 2020* | | |
| Cell line (*Homo sapiens*) | Human: HEK293T dMLLc7 | Generated in-house *Miyamoto et al., 2020* | | |
| Cell line (*Homo sapiens*) | Human: HEK293TL | Generated in-house *Okuda et al., 2015* | | |
| Strain, strain background (*Mus musculus*) | Mouse: C57BL/6J | CLEA Japan | | |

