## [Decision Letter]

**Acceptance summary:**

This is a quite comprehensive study demonstrating that various MLL fusions transform hematopoietic progenitors via HBO1-MLL interaction. The additional data provided in the revised manuscript have addressed most prior questions. This study adds mechanistic depth into the important recent discovery of HBO1 functions in MLL-fusion leukemias and provides potentially new therapeutic approach.

**Decision letter after peer review:**

Thank you for submitting your article "HBO1-MLL interaction promotes AF4/ENL/P-TEFb-mediated leukemogenesis" for consideration by *eLife*. Your article has been reviewed by 3 peer reviewers, including Xiaobing Shi as Reviewing Editor and Reviewer #1, and the evaluation has been overseen by Kevin Struhl as the Senior Editor.

Essential revisions:

The manuscript by Takahashi et al., describes the interaction between MLL fusion proteins with HBO1 and its role in leukemogenesis. Myeloid progenitor transformation assays using various MLL fusion proteins reveal that MLL fusion proteins requires the TRX2 domain of MLL for effective leukemic transformation. IP-MS identifies HBO1 as a bona fide binding partner of the MLL TRX2 domain. ChIP-seq experiments in various cancer cell lines indicate overlap between MLL and some of the HBO complex subunits, however this does not establish recruitment at these sites. ChIP-qPCR at a few MLL-fusion target genes with MLL depletion supports the recruitment hypothesis somewhat although with only mixed and modest effect sizes. At this point the paper shifts to a seemingly distinct line of inquiry, which is not closely related to the HBO1-TRX2 story to the first three figures. The new direction examines the ELL fusion partner in some detail using similar fusion protein chimeras, but a portion of Figure 4, is largely confirmatory of previously established findings about the critical regions of ELL for transformation and its AF4/EAF1 partners, adding only that portions of the MLL fusion protein are dispensable, provided that they are replaced with the PWWP of LEDGF. Further experiments define potentially two distinct complexes that have already been characterized being recruited by ELL, although there is overlap here again with their previous studies, and the results are a little hard to interpret. The last figure in the paper tackles the seemingly unrelated Nup98-HBO1 fusion, a rare patient mutation-they demonstrate a requirement for MLL for viability of hematopoietic progenitors transformed by this fusion, connecting back to the TRX2 interaction, and show that Menin inhibitors slow growth.

Overall, this is a quite comprehensive study demonstrating that various MLL fusions transform hematopoietic progenitors via HBO1-MLL interaction. The quality of the science fairly high (with a few noted exceptions that can be addressed), but the lack of clarity in presentation actively obscures really nice and seemingly important elements of this work. It also stops short of really demonstrating that the MLL-TRX2 motif recruits HBO1 in some direct way, which would be of high impact if it could be more clearly molecularly defined.

1. The most significant impact of this manuscript is the identification of the TRX2 region of the MLL-N protein as the major point of contact (perhaps not direct), to the HBO1 complex, which adds mechanistic depth to the important recent discovery that the MLL-fusion leukemias rely on HBO1 function (MacPherson et al., Nature 2020). However, it is not clear whether this interaction is direct, or bridged by some, as yet unidentified factor including DNA. It is also important to define the point of contact on the HBO1 side (even which subunit).

2. The current evidence does not conclusively demonstrate that The MLL-HBO1 interaction is the mechanism of HBO1 recruitment. It would be helpful to do a ChIP experiment in the MLL null background rescuing with +/- TRX2 to demonstrate that this "domain" is responsible for the recruitment-- even more ideal, but perhaps beyond the scope would be a point-mutant that disrupts direct contact to the HBO1 complex.

3. A large number of ChIP-seq and RNA-seq are performed. However, it is not clear how many repeats were performed. This need to be clearly stated, and biological repeats are needed for at least some of the essential ones, such as MLL1, HBO1 and H3K14ac, the mark deposited by HBO1. It is also important to describe clearly the source and specificity of the antibodies used for ChIP.

4. It seems that characterization of the AF4 and EAF1 associated complexes (in Figures 4-6) is unrelated to the main topic of this paper. This part is essentially a reinterpretation of prior results from Okuda, JCI 2017. This portion of the prior paper should be corrected there.

---

## [Author Response]

Essential revisons:1. The most significant impact of this manuscript is the identification of the TRX2 region of the MLL-N protein as the major point of contact (perhaps not direct), to the HBO1 complex, which adds mechanistic depth to the important recent discovery that the MLL-fusion leukemias rely on HBO1 function (MacPherson et al., Nature 2020). However, it is not clear whether this interaction is direct, or bridged by some, as yet unidentified factor including DNA. It is also important to define the point of contact on the HBO1 side (even which subunit).

To determine the major points of contact, we performed an immunoprecipitation followed by western blotting (IP-WB) analysis of the chromatin fractions from HEK293T cells transiently expressing various components of the HBO1 complex and the MLL bait protein (Figure 8). When each HBO1 complex component was co-expressed with the MLL bait protein, PHF16 and ING5 were weakly detected in the coprecipitates of the MLL bait protein, whereas HBO1 and MEAF6 were not (Figure 8A), suggesting that PHF16 and ING5 provide major points of contact for MLL. When all of the HBO complex components were co-expressed with the MLL bait protein, a very strong association was observed (Figure 8A, see fMLL 869/1124 + HBO1 complex), suggesting that MLL associates with the HBO1 complex through multiple contacts. Next, we omitted one component each from the HBO1 complex and tested which component provides the most significant contact for MLL. Omission of PHF16 or ING5 resulted in a drastic decrease of MLL-HBO1 complex association (Figure 8B), supporting the notion that PHF16 and ING5 provide major points of contact. Indeed, NUP98-ING5 fusion co-IPed MLL more efficiently compared to NUP98-HBO1 and NUP98-MEAF6 (Figure 7D). To further narrow down the structures responsible for MLL-HBO complex interaction, we generated a series of ING5 mutants and tested their association with MLL. Deleting the ING domain from ING5 resulted in the loss of MLL interaction (Figure 8C). In addition, mutations in the aromatic cage of the PHD finger of ING5, which is critical for methylation-specific association with Histone H3K4, also resulted in the loss of MLL interaction. This suggests that ING5 needs to be bound to methylated histone H3K4 and possibly mediates direct interaction with MLL via its ING domain. These results indicate that MLL-HBO1 complex interaction is mediated by multiple points of contact mainly mediated by PHF16 and ING5 in the chromatin-bound form. These new observations/notions have been added to the revised manuscript. Overall, our results indicated that MLL-HBO1 complex formation is highly context-dependent, which preferentially occurs on chromatin as a fully assembled HBO1 complex. As DNAseI treatment did not hinder the interaction, we believe that DNA is not required for MLL-HBO1 interaction (Figure 2—figure supplement 2B).

2. The current evidence does not conclusively demonstrate that The MLL-HBO1 interaction is the mechanism of HBO1 recruitment. It would be helpful to do a ChIP experiment in the MLL null background rescuing with +/- TRX2 to demonstrate that this "domain" is responsible for the recruitment-- even more ideal, but perhaps beyond the scope would be a point-mutant that disrupts direct contact to the HBO1 complex.

We have provided ChIP-qPCR analysis data of HEK293T cells lacking MLL (Figure 3F). We performed ChIP for HBO1 using 3 biological replicates and observed a consistent decrease in the HBO1 ChIP signals at the MYC promoter, supporting the idea that MLL-HBO1 interaction is the mechanism of HBO1 recruitment, at least partially, at the MYC promoter. However, it is also true that HBO1 is recruited to other non-MYC promoters such as RPL13A in the absence of MLL, indicating that HBO1 is recruited to the promoter in non-MLL-mediated manners as well. H3K4me3 marks were substantially decreased at the MYC promoter but not at the RPL13A promoter in MLL-deficient cells. This suggested that MLL-mediated H3K4 methylation is required for ING4/5-mediated HBO1 complex recruitment. As advised by the reviewer, we examined the ability of MLL mutants, with or without THD2 (Formally termed as TRX2 domain), to recruit HBO1 in MLL null background. We did not observe any substantial increase of HBO1 recruitment by the THD2-containing MLL mutant (Figure 3—figure supplement 1C), indicating that THD2 is not sufficient for HBO1 recruitment but requires the pre-existing H3K4me3 marks. We think that the MLL-HBO1 interaction enhances the recruitment of HBO1 to certain promoters such as MYC, while the HBO1 can be recruited to promoters in other ways. We speculate that gene promoters which depend on the AEP/SL1-mediated transcriptional pathway are heavily dependent on the MLL-HBO1 complex. In accord, the MYC promoter shows a relatively robust decrease of the ChIP signals of AEP/SL1 (i.e., ENL, CCNT1, CDK9 and TAF1C) in MLL knockout cells (Figure 3 and Figure 3—figure supplement1B). In the revised manuscript, we have added these new findings and modified the text to avoid giving the misleading impression that MLL is the sole element whose presence is sufficient for the HBO1 complex recruitment.

3. A large number of ChIP-seq and RNA-seq are performed. However, it is not clear how many repeats were performed. This need to be clearly stated, and biological repeats are needed for at least some of the essential ones, such as MLL1, HBO1 and H3K14ac, the mark deposited by HBO1. It is also important to describe clearly the source and specificity of the antibodies used for ChIP.

We have performed ChIP-seq analyses of MLL (MLL1) and HBO1complex in duplicates and deposited in the public archive. ChIP-seq data of H3K14ac was not presented in the initial manuscript. The HBO1/BRPF2 complex was shown to be responsible for the global acetylation of H3K14ac in the hematopoietic cells (1). However, it is considered that the preferred histone modifications provided by the HBO1/PHF16 complex are histone H4K5/8/12 acetylations rather than H3K14/23 acetylations (2). We apologize for indicating that the HBO1/PHF16 complex provides H3K14ac marks in the cartoon in the previous manuscript, which was incorrect. Thus, PHF16-, MEAF6-, and ING4-ChIP analyses were performed to more clearly demonstrate the specific localization of the HBO1 complex and the results have been presented. Details of the antibodies used for ChIP experiments are described in the revised manuscript in Key resource table.

4. It seems that characterization of the AF4 and EAF1 associated complexes (in Figures 4-6) is unrelated to the main topic of this paper. This part is essentially a reinterpretation of prior results from Okuda, JCI 2017. This portion of the prior paper should be corrected there.

The main focus of this study was understanding the role of the HBO1 complex in MLL-mediated transactivation pathways. The structure/function analysis of MLL fusion proteins demonstrated that MLL-ELL is highly dependent on the HBO1-mediated function in leukemic transformation. Hence, it was important to clarify the mechanism of gene activation by MLL-ELL in this paper to understand why HBO1 association is critically required for MLL-ELL-mediated transformation. Because MLL-ELL associates with AEP similarly to other major MLL fusions such as MLL-AF4 and MLL-ENL, it was speculated that MLL-ELL also activates its target genes via AEP. However, ELL associates with EAF family proteins and MLL-EAF also has transforming ability (3), which contradicts this idea. Thus, EAF1-mediated function could be more important for MLL-ELL-mediated transformation. To dissect the molecular mechanism of transformation, we generated a point mutant that selectively impaired ELL-EAF interaction and demonstrated that EAF1-association is dispensable for MLL-ELL-mediated transformation (Figure 6), thereby indicating that MLL-ELL transforms via AEP-mediated functions, which demands HBO1-mediated functions. We also showed that the presence of THD2 enhances ELL-AEP association to further suggest that one of the roles for the HBO1 complex is to enhance the association of ELL with AEP. These findings are not reinterpretations of our prior results and are relevant to the main topic of this paper, and therefore we have retained them in the revised manuscript.